# One-Line-of-Code Data Mollification Improves Optimization of Likelihood-based Generative Models

**Ba-Hien Tran**
Department of Data Science
EURECOM, France
`ba-hien.tran@eurecom.fr`

**Giulio Franzese**
Department of Data Science
EURECOM, France
`giulio.franzese@eurecom.fr`

**Pietro Michiardi**
Department of Data Science
EURECOM, France
`pietro.michiardi@eurecom.fr`

**Maurizio Filippone**
Department of Data Science
EURECOM, France
`maurizio.filippone@eurecom.fr`

## Abstract

Generative Models (GMs) have attracted considerable attention due to their tremendous success in various domains, such as computer vision where they are capable to generate impressive realistic-looking images. Likelihood-based GMs are attractive due to the possibility to generate new data by a single model evaluation. However, they typically achieve lower sample quality compared to state-of-the-art score-based Diffusion Models (DMs). This paper provides a significant step in the direction of addressing this limitation. The idea is to borrow one of the strengths of score-based DMs, which is the ability to perform accurate density estimation in low-density regions and to address manifold overfitting by means of data mollification. We propose a view of data mollification within likelihood-based GMs as a continuation method, whereby the optimization objective smoothly transitions from simple-to-optimize to the original target. Crucially, data mollification can be implemented by adding one line of code in the optimization loop, and we demonstrate that this provides a boost in generation quality of likelihood-based GMs, without computational overheads. We report results on real-world image data sets and UCI benchmarks with popular likelihood-based GMs, including variants of variational autoencoders and normalizing flows, showing large improvements in FID score and density estimation.

## 1   Introduction

Generative Models (GMs) have attracted considerable attention recently due to their tremendous success in various domains, such as computer vision, graph generation, physics and reinforcement learning [see e.g., 37, 39, 75, 76, and references therein]. Given a set of data points, GMs attempt to characterize the distribution of such data so that it is then possible to draw new samples from this. Popular approaches include Variational Autoencoders (VAEs), Normalizing Flows (NFs), Generative Adversarial Networks (GANs), and score-based Diffusion Models (DMs).

In general, the goal of any GMs is similar to that of density estimation with the additional aim to do so by constructing a parametric mapping between an easy-to-sample-from distribution $p_s$ and the desired data distribution $p_{data}$. While different GMs approaches greatly differ in their optimization strategy and formulation, the underlying objectives share some similarity due to their relation to the optimal transport problem, defined as $\arg\min_\pi \int \|\mathbf{x} - \mathbf{y}\|^2 d\pi(\mathbf{x}, \mathbf{y})$. Here $\pi$ is constrained to belong to the set of joint distributions with marginals $p_s, p_{data}$, respectively [19, 42]. This unified

37th Conference on Neural Information Processing Systems (NeurIPS 2023).

perspective is explicitly investigated for GANs and VAEs [19] for example, whereas other works study NFs [50]. Similarly, DMs can be connected to Schrodinger Bridges [8], which solve the problem of *entropy-regularized* optimal transport [55]. Given that extensions of the regularized optimal transport case are available also for other generative models [65, 58], we should expect that, in principle, any technique should allow generation of samples with similar quality, provided it is properly tuned. However, this is not true in practice. The different formulations lead to a variety of properties associated with GMs, and pros and cons of each formulation can be understood through the so-called GM tri-lemma [84]. The three desirable properties of GMs are high sample quality, mode coverage, and fast sampling, and it has been argued that such goals are difficult to be satisfied simultaneously [84] .

The state-of-the-art is currently dominated by score-based DMs, due to their ability to achieve high sample quality and good mode coverage. However, generating new samples is computationally expensive due to the need to simulate stochastic differential equations. Likelihood-based GMs are complementary, in that they achieve lower sample quality, but sampling requires one model evaluation per sample and it is therefore extremely fast. While some attempts have been made to bridge the gap by combining GANs with DMs [84] or training GANs with diffusions [82], these still require careful engineering of architectures and training schedules. The observation that all GMs share a common underlying objective indicates that we should look at what makes DMs successful at optimizing their objective. Then, the question we address in this paper is: can we borrow the strengths of score-based DMs to improve likelihood-based GMs, without paying the price of costly sample generation?

One distinctive element of score-based DMs is data mollification, which is typically achieved by adding Gaussian noise [69] or, in the context of image data sets, by blurring [61]. A large body of evidence points to the *manifold hypothesis* [63], which states that the intrinsic dimensionality of image data sets is much lower than the dimensionality of their input. Density estimation in this context is particularly difficult because of the degeneracy of the likelihood for any density concentrated on the manifold where data lies [43]. Under the manifold hypothesis, or even when the target density is multi-modal, the Lipschitz constant of GMs has to be large, but regularization, which is necessary for robustness, is antagonist to this objective [64, 9]. As we will study in detail in this paper, the process of data mollification gracefully guides the optimization mitigating manifold overfitting and enabling a desirable tail behavior, yielding accurate density estimation in low-density regions. In likelihood-based GMs, data mollification corresponds to some form of simplification of the optimization objective. This type of approach, where the level of data mollification is annealed throughout training, can be seen as a continuation method [83, 47], which is a popular technique in the optimization literature to reach better optima.

Strictly speaking, data mollification in score-based DMs and likelihood-based GMs are slightly different. In the latter, the amount of noise injected in the data is continuously annealed throughout training. At the beginning, the equivalent loss landscape seen by the optimizer is much smoother, due to the heavy perturbation of the data, and a continuous reduction of the noise level allows optimization to be gracefully guided until the point where the level of noise is zero [83, 47]. DMs, instead, are trained at each step of the optimization process by considering **all** noise levels simultaneously, where complex amortization procedures, such as self-attention [70], allow the model to efficiently share parameters across different perturbation levels. It is also worth mentioning that score-based DMs possess another distinctive feature in that they perform gradient-based density estimation [69, 28]. It has been conjectured that this can be helpful to avoid manifold overfitting by allowing for the modeling of complex densities while keeping the Lipschitz constant of score networks low [64]. In this work, we attempt to verify the hypothesis that data mollification is heavily responsible for the success of score-based DMs. We do so by proposing data mollification for likelihood-based GMs, and provide theoretical arguments and experimental evidence that data mollification consistently improves their optimization. Crucially, this strategy yields better sample quality and it is extremely easy to implement, as it requires adding very little code to any existing optimization loop.

We consider a large set of experiments involving VAEs and NFs and some popular image data sets. These provide a challenging test for likelihood-based GMs due to the large dimensionality of the input space and to the fact that density estimation needs to deal with data lying on manifolds. The results show systematic, and in some cases dramatic, improvements in sample quality, indicating that this is a simple and effective strategy to improve optimization of likelihood-based GMs models. The paper is organized as follows: in § 2 we illustrate the challenges associated with generative modeling when data points lie on a manifold, particularly with density estimation in low-density regions and

manifold overfitting; in § 3 we propose data mollification to address these challenges; § 4 reports the experiments with a discussion of the limitations and the broader impact, while § 5 presents related works, and § 6 concludes the paper.

## 2   Challenges in Training Deep Generative Models

We are interested in unsupervised learning, and in particular on the task of density estimation. Given a dataset $\mathcal{D}$ consisting of $N$ i.i.d samples $\mathcal{D} \triangleq \{\mathbf{x}_i\}_{i=1}^N$ with $\mathbf{x}_i \in \mathbb{R}^D$, we aim to estimate the unknown continuous generating distribution $p_{\text{data}}(\mathbf{x})$. In order to do so, we introduce a model $p_{\boldsymbol{\theta}}(\mathbf{x})$ with parameters $\boldsymbol{\theta}$ and attempt to estimate $\boldsymbol{\theta}$ based on the dataset $\mathcal{D}$. A common approach to estimate $\boldsymbol{\theta}$ is to maximize the likelihood of the data, which is equivalent to minimizing the following objective:

$$\mathcal{L}(\boldsymbol{\theta}) \triangleq -\mathbb{E}_{p_{\text{data}}(\mathbf{x})} \left[ \log p_{\boldsymbol{\theta}}(\mathbf{x}) \right]. \tag{1}$$

There are several approaches to parameterize the generative model $p_{\boldsymbol{\theta}}(\mathbf{x})$. In this work, we focus on two widely used likelihood-based Generative Models (GMs), which are Normalizing Flows (NFs) [52, 39] and Variational Autoencoders (VAEs) [36, 59]. Although NFs and VAEs are among the most popular deep GMs, they are characterized by a lower sample quality compared to GANs and score-based DMs. In this section, we present two major reasons behind this issue by relying on the manifold hypothesis.

### 2.1   The Manifold Hypothesis and Density Estimation in Low-Density Regions

The manifold hypothesis is a fundamental concept in manifold learning [63, 73, 1] stating that real-world high-dimensional data tend to lie on a manifold $\mathcal{M}$ characterized by a much lower dimensionality compared to the one of the input space (ambient dimensionality) [48]. This has been verified theoretically and empirically for many applications and datasets [51, 48, 57, 72]. For example, [57] report extensive evidence that natural image datasets have indeed very low intrinsic dimension relative to the high number of pixels in the images.

The manifold hypothesis suggests that density estimation in the input space is challenging and ill-posed. In particular, data points on the manifold should be associated with high density, while points outside the manifold should be considered as lying in regions of nearly zero density [45]. This implies that the target density in the input space should be characterized by high Lipschitz constants. The fact that data is scarce in regions of low density makes it difficult to expect that models can yield accurate density estimation around the tails. These pose significant challenges for the training of deep GMs [9, 45, 69]. Recently, diffusion models [69, 25, 70] have demonstrated the ability to mitigate this problem by gradually transforming a Gaussian distribution, whose support spans the full input space, into the data distribution. This observation induces us to hypothesize that the

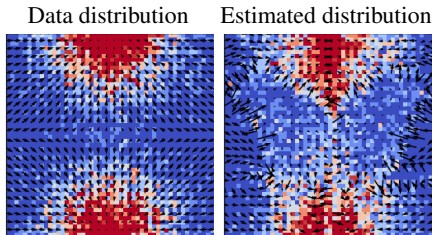

Data distribution      Estimated distribution

**Figure 1:** **Left:** Histogram of samples from data distribution $p_{\text{data}}(\mathbf{x})$ and its true scores $\nabla_{\mathbf{x}} \log p_{\text{data}}(\mathbf{x})$; **Right:** Histogram of of samples from the estimated distribution $p_{\boldsymbol{\theta}}(\mathbf{x})$ and its scores $\nabla_{\mathbf{x}} \log p_{\boldsymbol{\theta}}(\mathbf{x})$. In the low density regions, the model is unable to capture the true density and scores.

data mollification mechanism in score-based DMs is responsible for superior density estimation in low-density regions.

To demonstrate the challenges associated with accurate estimation in low-density regions, we consider a toy experiment where we use a REAL-NVP flow [14] to model a two-dimensional mixture of Gaussians, which is a difficult test for NFs in general. Details on this experiment are provided in the Appendix D. Fig. 1 depicts the true and estimated densities, and their corresponding scores, which are the gradient of the log-density function with respect to the data [28]. Note that the use of "score" here is slightly different from that from traditional statistics where score usually refers to the gradient of the log-likelihood with respect to model parameters. As it can be seen in the figure, in regions of low data density, $p_{\boldsymbol{\theta}}(\mathbf{x})$ is completely unable to model the true density and scores. This problem is due to the lack of data samples in these regions and may be more problematic under the manifold hypothesis and for high-dimensional data such as images. In § 4, we will demonstrate how it is possible to considerably mitigate this issue by means of data mollification.

## 2.2 Manifold Overfitting

The manifold hypothesis suggests that overfitting on a manifold can occur when the model $p_{\boldsymbol{\theta}}(\mathbf{x})$ assigns an arbitrarily large likelihood in the vicinity of the manifold, even if the distribution does not accurately capture the true distribution $p_{\text{data}}(\mathbf{x})$ [10, 43]. This issue is illustrated in Fig. 2 of [43] and it will be highlighted in our experiment (§ 4.1), where the true data distribution $p_{\text{data}}(\mathbf{x})$ is supported on a one-dimensional curve manifold $\mathcal{M}$ in two-dimensional space $\mathbb{R}^2$. Even when the model distribution $p_{\boldsymbol{\theta}}(\mathbf{x})$ poorly approximates $p_{\text{data}}(\mathbf{x})$, it may reach a high likelihood value by concentrating the density around the correct manifold $\mathcal{M}$. If $p_{\boldsymbol{\theta}}(\mathbf{x})$ is flexible enough, any density defined on $\mathcal{M}$ may achieve infinite likelihood and this might be an obstacle for retrieving $p_{\text{data}}(\mathbf{x})$.

A theoretical formalization of the problem of manifold overfitting appears in [43] and it is based on the concept of Riemannian measure [56]. The Riemannian measure on manifolds holds an analogous role to that of the Lebesgue measure on Euclidean spaces. To begin, we establish the concept of smoothness for a probability measure on a manifold.

**Definition 1.** *Let $\mathcal{M}$ be a finite-dimensional manifold, and $p$ be a probability measure on $\mathcal{M}$. Let $\mathfrak{g}$ be a Riemannian metric on $\mathcal{M}$ and $\mu_{\mathcal{M}}^{(\mathfrak{g})}$ the corresponding Riemannian measure. We say that $p$ is smooth if $p \ll \mu_{\mathcal{M}}^{(\mathfrak{g})}$ and it admits a continuous density $p : \mathcal{M} \to \mathbb{R}_{>0}$ with respect to $\mu_{\mathcal{M}}^{(\mathfrak{g})}$.*

We now report Theorem 1 from [43] followed by a discussion on its implications for our work.

**Theorem 1.** *(Gabriel Loaiza-Ganem et al. [43]). Let $\mathcal{M} \subset \mathbb{R}^D$ be an analytic $d$-dimensional embedded submanifold of $\mathbb{R}^d$ with $d < D$, $\mu_D$ is the Lebesgue measure on $\mathbb{R}^D$, and $p^\dagger$ a smooth probability measure on $\mathcal{M}$. Then there exists a sequence of probability measures $\{p_{\boldsymbol{\theta}}^{(t)}\}_{t=0}^{\infty}$ on $\mathbb{R}^D$ such that:*

1. *$p_{\boldsymbol{\theta}}^{(t)} \to p^\dagger$ as $t \to \infty$.*

2. *$\forall t \geq 0, p_{\boldsymbol{\theta}}^{(t)} \ll \mu_D$ and $p_{\boldsymbol{\theta}}^{(t)}$ admits a density $p_{\boldsymbol{\theta}}^{(t)} : \mathbb{R}^D \to \mathbb{R}_{>0}$ with respect to $\mu_D$ such that:*
    - *(a) $\lim_{t \to \infty} p_{\boldsymbol{\theta}}^{(t)}(\mathbf{x}) = \infty, \forall \mathbf{x} \in \mathcal{M}$.*
    - *(b) $\lim_{t \to \infty} p_{\boldsymbol{\theta}}^{(t)}(\mathbf{x}) = 0, \forall \mathbf{x} \notin cl(\mathcal{M})$, where $cl(\cdot)$ denotes closure in $\mathbb{R}^D$.*

Theorem 1 holds for any smooth probability measure supported in $\mathcal{M}$. This is an important point because this includes the desired $p_{\text{data}}$, provided that this is smooth too. The key message in [43] is that, a-priori, there is no reason to expect that for a likelihood-based model to converge to $p_{\text{data}}$ out of all the possible $p^\dagger$. Their proof is based on convolving $p^\dagger$ with a Gaussian kernel with variance $\sigma_t^2$ that decreases to zero as $t \to \infty$, and then verify that the stated properties of $p_{\boldsymbol{\theta}}^{(t)}$ hold. Our analysis, while relying on the same technical tools, is instead constructive in explaining why the proposed data mollification allows us to avoid manifold overfitting. The idea is as follows: at time step $t = 0$, we select the desired $p_{\text{data}}$ convolved with a Gaussian kernel with a large, but finite, variance $\sigma^2(0)$ as the target distribution for the optimization. Optimization is performed and $p_{\boldsymbol{\theta}}^{(0)}$ targets this distribution, without any manifold overfitting issues, since the dimensionality of the corrupted data is non-degenerate. At the second step, the target distribution is obtained by convolving $p_{\text{data}}$ with the kernel with variance $\sigma^2(1) < \sigma^2(0)$, and again manifold overfitting is avoided. By iteratively repeating this procedure, we can reach the point where we are matching a distribution convolved with an arbitrarily small variance $\sigma^2(t)$, without ever experiencing manifold overfitting. When removing the last bit of perturbation we fall back to the case where we experience manifold overfitting. However, when we operate in a stochastic setting, which is the typical scenario for the GMs considered here, we avoid ending up in solutions for which the density is degenerate and with support which is exactly the data manifold. Another way to avoid instabilities is to adopt gradient clipping. Note that, as mentioned in [43] and verified by ourselves in earlier investigations, a small constant amount of noise does not provide any particular benefits over the original scheme, whereas gradually reducing the level of data mollification improves optimization dramatically.

## 2.3 Data Mollification as a Continuation Method

We can view the proposed data mollification approach as a continuation method [83, 47]. Starting from the target objective function, which in our case is $\mathcal{L}(\boldsymbol{\theta})$ in Eq. 4 (or a lower bound in the case of VAEs), we construct a family of functions $\mathcal{H}(\boldsymbol{\theta}, \gamma)$ parameterized by an auxiliary variable

$\gamma \in [0, 1]$ so that $\mathcal{H}(\boldsymbol{\theta}, 0) = \mathcal{L}(\boldsymbol{\theta})$. The objective functions $\mathcal{H}(\boldsymbol{\theta}, \gamma)$ are defined so that the higher $\gamma$ the easier is to perform optimization. In our case, when $\gamma = 1$ we operate under a simple regime where we target a Gaussian distribution, and likelihood-based GMs can model these rather easily. By annealing $\gamma$ from 1 to 0 with a given schedule, the sequence of optimization problems with objective $\mathcal{H}(\boldsymbol{\theta}, \gamma)$ is increasinly more complex to the point where we target $\mathcal{L}(\boldsymbol{\theta})$. In essence, the proposed data mollification approach can be seen as a good initialization method, as the annealing procedure introduces a memory effect in the optimization process, which is beneficial in order to obtain better optima.

## 3    Generative Models with Data Mollification

Motivated by the aforementioned problems with density estimation in low-density regions and manifold overfitting, we propose a simple yet effective approach to improve likelihood-based GMs. Our method involves mollifying data using Gaussian noise, gradually reducing its variance, until recovering the original data distribution $p_{\text{data}}(\mathbf{x})$. This mollification procedure is similar to the reverse process of diffusion models, where a prior noise distribution is smoothly transformed into the data distribution [69, 25, 70]. As already mentioned, data mollification alleviates the problem of manifold overfitting and it induces a memory effect in the optimization which improves density estimation in regions of low density.

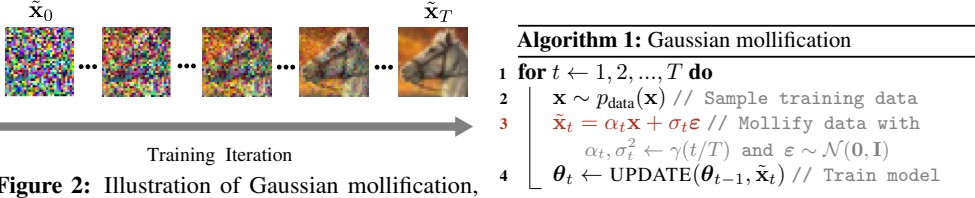

**Figure 2:** Illustration of Gaussian mollification, where $\tilde{\mathbf{x}}_t$ is the mollified data at iteration $t$.

**Algorithm 1:** Gaussian mollification
1  **for** $t \leftarrow 1, 2, ..., T$ **do**
2     $\mathbf{x} \sim p_{\text{data}}(\mathbf{x})$ // Sample training data
3     $\tilde{\mathbf{x}}_t = \alpha_t \mathbf{x} + \sigma_t \boldsymbol{\varepsilon}$ // Mollify data with
        $\alpha_t, \sigma_t^2 \leftarrow \gamma(t/T)$ and $\boldsymbol{\varepsilon} \sim \mathcal{N}(\mathbf{0}, \mathbf{I})$
4     $\boldsymbol{\theta}_t \leftarrow \text{UPDATE}(\boldsymbol{\theta}_{t-1}, \tilde{\mathbf{x}}_t)$ // Train model

**Gaussian Mollification.**    Given that we train the model $p_{\boldsymbol{\theta}}(\mathbf{x})$ for $T$ iterations, we can create a sequence of progressively less smoothed versions of the original data $\mathbf{x}$, which we refer to as mollified data $\tilde{\mathbf{x}}_t$. Here, $t$ ranges from $t = 0$ (the most mollified) to $t = T$ (the least mollified). For any $t \in [0, T]$, the distribution of the mollified data $\tilde{\mathbf{x}}_t$, conditioned on $\mathbf{x}$, is given as follows:

$$q(\tilde{\mathbf{x}}_t \mid \mathbf{x}) = \mathcal{N}(\tilde{\mathbf{x}}_t; \alpha_t \mathbf{x}, \sigma_t^2 \mathbf{I}), \tag{2}$$

where $\alpha_t$ and $\sigma_t^2$ are are positive scalar-valued functions of $t$. In addition, we define the signal-to-noise ratio $\text{SNR}(t) = \alpha_t^2 / \sigma_t^2$. and we assume that it monotonically increases with $t$, i.e., $\text{SNR}(t) \leq \text{SNR}(t + 1)$ for all $t \in [0, T - 1]$. In other words, the mollified data $\tilde{\mathbf{x}}_t$ is progressively less smoothed as $t$ increases. In this work, we adopt the *variance-preserving* formulation used for diffusion models [67, 25, 32], where $\alpha_t = \sqrt{1 - \sigma_t^2}$ and $\sigma_t^2 = \gamma(t/T)$. Here, $\gamma(\cdot)$ is a monotonically decreasing function from 1 to 0 that controls the rate of mollification. Intuitively, this procedure involves gradually transforming an identity-covariance Gaussian distribution into the distribution of the data. Algorithm 1 summarizes the proposed Gaussian mollification procedure, where the red line indicates a simple additional step required to mollify data compared with vanilla training using the true data distribution.

**Noise schedule.**    The choice of the noise schedule $\gamma(\cdot)$ has an impact on the performance of the final model. In this work, we follow common practice in designing the noise schedule based on the literature of score-based DMs [49, 26, 7]. In particular, we adopt a sigmoid schedule [31], which has recently been shown to be more effective in practice compared to other choices such as linear [25] or cosine schedules [49].

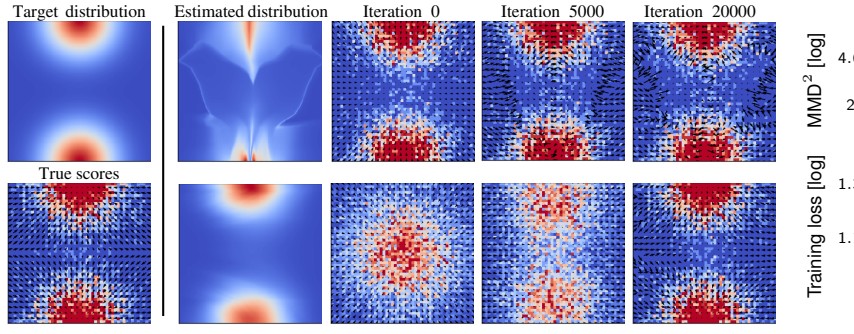

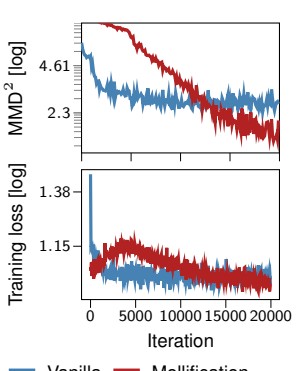

**Figure 4:** The first column shows the target distribution and the true scores. The second column depicts the estimated distributions of the Gaussian Mixture Model (GMM) , which yield MMD$^2$ of 15.5 and 2.5 for the vanilla (top) and mollification (bottom) training, respectively. The remaining columns show histogram of samples from the true (**top row**) and mollified data (**bottom row**), and estimated scores.

**Figure 5:** The learning curves of the GMM experiments.

The sigmoid schedule $\gamma(t/T)$ [31] is defined through the sigmoid function:

$$\text{sigmoid}\left(-\frac{t/T}{\tau}\right), \qquad (3)$$

where $\tau$ is a temperature hyper-parameter. This function is then scaled and shifted to ensure that $\gamma(0) = 1$ and $\gamma(1) = 0$. We encourage the reader to check the implementation of this schedule, available in Appendix C. Fig. 3 illustrates the sigmoid schedule and the corresponding $\log(\text{SNR})$ with different values of $\tau$. We use a default temperature of $0.7$ as it demonstrates consistently good results in our experiments.

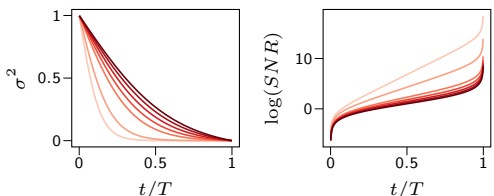

**Figure 3:** Illustration of sigmoid schedule and the corresponding $\log(\text{SNR})$. The temperature values from $0.2$ to $0.9$ are progressively shaded, with the lighter shade corresponding to lower temperatures.

## 4 Experiments

In this section, we demonstrate emprically the effectiveness of our proposal through a wide range of experiments on synthetic data, and some popular real-world tabular and image data sets. Appendix D contains a detailed description of each experiment to guarantee reproducibility.

### 4.1 2D Synthetic Data Sets

We begin our experimental campaign with two 2D synthetic data sets. The two-dimensional nature of these data sets allows us to demonstrate the effectiveness of Gaussian mollification in mitigating the challenges associated with density estimation in low-density regions and manifold overfitting. Here, we consider $p_{\boldsymbol{\theta}}(\mathbf{x})$ to be a REAL-NVP flow [14], which comprises five coupling bijections, each consisting of a two-hidden layer multilayer perceptron (MLP). To assess the capability of $p_{\boldsymbol{\theta}}(\mathbf{x})$ to recover the true data distribution, we use Maximum Mean Discrepancy (MMD) [21] with a radial basis function (RBF) kernel on a held-out set. In these experiments, we employ the Gaussian mollification strategy presented in the previous section and compare the estimated density with the *vanilla* approach where we use the original training data without any mollification.

**Mixture of Gaussians.** First, we consider a target distribution that is a mixture of two Gaussians, as depicted in Fig. 4. As discussed in § 2.1, the vanilla training procedure fails to accurately estimate the true data distribution and scores, particularly in the low-density regions. The estimated densities and the mollified data during the training are depicted in Fig. 4. Initially, the mollification process considers a simpler coarse-grained version of the target density, which is easy to model. This is demonstrated by the low training loss at the beginning of the optimization, as depicted in Fig. 5. Subsequently, the method gradually reduces the level of noise allowing for a progressive refinement of the estimated versions of the target density. This process uses the solution from one level of

mollification as a means to guiding optimization for the next. As a result, Gaussian mollification facilitates the recovery of the modes and enables effective density estimation in low-density regions. The vanilla training procedure, instead, produces a poor estimate of the target density, as evidenced by the trace-plot of the $\text{MMD}^2$ metric in Fig. 5 and the visualization of the scores in Fig. 4.

**Von Mises distribution.** We proceed with an investigation of the von Mises distribution on the unit circle, as depicted in Fig. 6, with the aim of highlighting the issue of manifold overfitting [43]. In this experiment, the data lies on a one-dimensional manifold embedded in a two-dimensional space. The vanilla training procedure fails to approximate the target density effectively, as evidenced by the qualitative results and the substantially high value of $\text{MMD}^2$ ($\approx 383.44$) shown in Fig. 6. In contrast, Gaussian mollification gradually guides the estimated density towards the target, as depicted in Fig. 6, leading to a significantly lower $\text{MMD}^2$ ($\approx 6.13$). Additionally, the mollification approach enables the estimated model not only to accurately learn the manifold but also to capture the mode of the density correctly.

## 4.2 Image Experiments

**Table 1:** FID score on CIFAR10 and CELEBA dataset (*lower is better*). The small colored numbers indicate improvement or degration of the mollification training compared to the vanilla training.

| MODEL | CIFAR10 | | | CELEBA | | |
|---|---|---|---|---|---|---|
| | VANILLA | GAUSSIAN MOLLIFCATION | BLURRING MOLLIFICATION | VANILLA | GAUSSIAN MOLLIFICATION | BLURRING MOLLIFICATOIN |
| REAL-NVP [14] | 131.15 | 121.75 ↓ 7.17% | 120.88 ↓ 7.83% | 81.25 | 79.68 ↓ 1.93% | 85.40 ↑ 5.11% |
| GLOW [34] | 74.62 | 64.87 ↓ 13.07% | 66.70 ↓ 10.61% | 97.59 | 70.91 ↓ 27.34% | 74.74 ↓ 23.41% |
| VAE[36] | 191.98 | 155.13 ↓ 19.19% | 175.40 ↓ 8.64% | 80.19 | 72.97 ↓ 9.00% | 77.29 ↓ 3.62% |
| VAE-IAF [35] | 193.58 | 156.39 ↓ 19.21% | 162.27 ↓ 16.17% | 80.34 | 73.56 ↓ 8.44% | 75.67 ↓ 5.81% |
| IWAE [4] | 183.04 | 146.70 ↓ 19.85% | 163.79 ↓ 10.52% | 78.25 | 71.38 ↓ 8.78% | 76.45 ↓ 2.30% |
| $\beta$-VAE [24] | 112.42 | 93.90 ↓ 16.47% | 101.30 ↓ 9.89% | 67.78 | 64.59 ↓ 4.71% | 67.08 ↓ 1.03% |
| HVAE [5] | 172.47 | 137.84 ↓ 20.08% | 147.15 ↓ 14.68% | 74.10 | 72.28 ↓ 2.46% | 77.54 ↑ 4.64% |

**Setup.** We evaluate our method on image generation tasks on CIFAR10 [40] and CELEBA 64 [41] datasets, using a diverse set of likelihood-based GMs. The evaluated models include the vanilla VAE [36], the $\beta$-VAE [24], and the VAE-IAF [35] which employs an expressive inverse autoregressive flow for the approximate posterior. To further obtain flexible approximations of the posterior of latent variables as well as a tight evidence lower bound (ELBO), we also select the Hamiltonian-VAE (HVAE) [5] and the importance weighted VAE (IWAE) [4]. For flow-based models, we consider the REAL-NVP [14] and GLOW [34] models in our benchmark. We found that further training the model on the original data after the mollification procedure leads to better performance. Hence, in our approach we apply data mollification during the first half of the optimization phase, and we continue optimize the model using the original data in the second half. Nevertheless, to ensure a fair comparison, we adopt identical settings for the vanilla and the proposed approaches, including random seed, optimizer, and the total number of iterations.

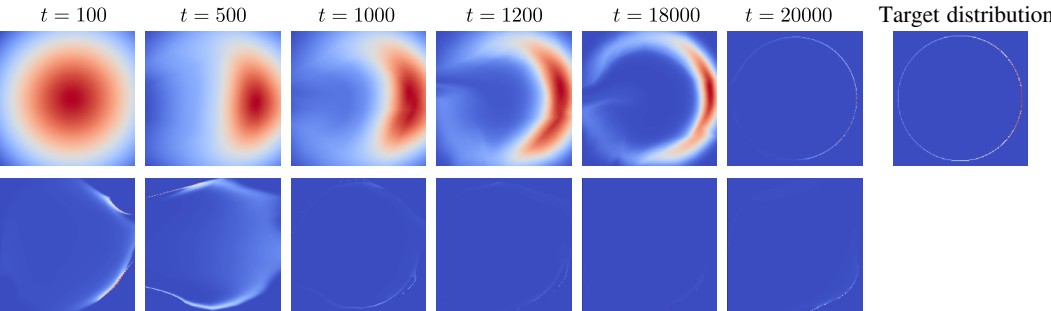

**Figure 6:** The progression of the estimated densities for the von Mises distribution from the vanilla (**bottom row**) and our mollification (**top row**) approaches.

**Blurring mollification.** Even though Gaussian mollification is motivated by the manifold hypothesis, it is not the only way to mollify the data. Indeed, Gaussian mollification does not take into account certain inductive biases that are inherent in natural images, including their multi-scale nature. Recently, [61, 27, 11] have proposed methods that incorporate these biases in diffusion-type generative models. Their approach involves stochastically reversing the heat equation, which is a partial differential equation (PDE) that can be used to erase fine-scale information when applied locally to the 2D plane of an image. During training, the model first learns the coarse-scale structure of the data, which is easier to learn, and then gradually learns the finer details. It is therefore interesting to assess whether this form of data mollification is effective in the context of this work compared to the addition of Gaussian noise. Note, however, that under the manifold hypothesis, this type of mollification produces the opposite effect to the addition of Gaussian noise in that at time $t = 0$ mollified images lie on a 1D manifold and they are gradually transformed to span the dimension of the data manifold; more details on blurring mollification can be found in Appendix B.

**Image generation.** We evaluate the quality of the generated images using the popular Fréchet Inception Distance (FID) score [22] computed on 50K samples from the trained model using the `pytorch-fid` [1] library. The results, reported in Table 1, indicate that the proposed data mollification consistently improves model performance compared to vanilla training across all datasets and models. Additionally, mollification through blurring, which is in line with recent results from diffusion models [61], is less effective than Gaussian mollification, although it still enhances the vanilla training in most cases. We also show intermediate samples in Fig. 8 illustrating the progression of samples from pure random noise or completely blurred images to high-quality images. Furthermore, we observe that Gaussian mollification leads to faster convergence of the FID score for VAE-based models, as shown in Fig. 7. We provide additional results in Appendix E. As a final experiment, we consider a recent large-scale VAE model for the CIFAR10 data set, which is a deep hierarchical VAE (NVAE) [78]. By applying Gaussian mollification without introducing any additional complexity, e.g., step-size annealing, we improve the FID score from 53.64 to 52.26.

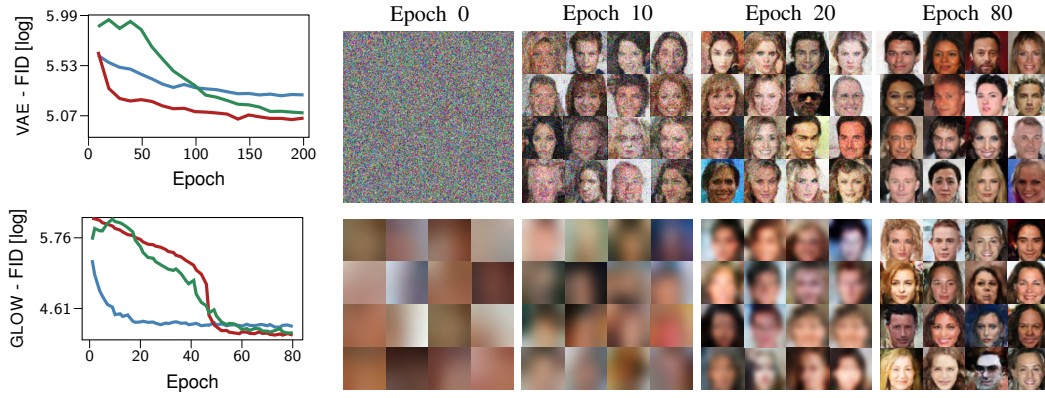

**Figure 7:** The progression of FID on CIFAR10 dataset.

**Figure 8:** Intermediate samples generated from REAL-NVP flows [14], which are trained on CELEBA dataset employed with Gaussian (**top row**) and blurring mollification (**bottom row**).

**Choice of noise schedule.** We ablate on the choice of noise schedule for Gaussian mollification. Along with the sigmoid schedule, we also consider the linear [25] and cosine [49] schedules, which are also popular for diffusion models. As shown in Table 2, our method consistently outperforms the vanilla approach under all noise schedules. We also observe that the sigmoid schedule consistently produced good results. Therefore, we chose to use the sigmoid schedule in all our experiments.

**Table 2:** FID score on CIFAR10 w.r.t. different choices of noise schedule.

| MODEL | VANILLA | SIGMOID | COSINE | LINEAR |
|---|---|---|---|---|
| REAL-NVP | 191.98 | 121.75 | 118.71 | 123.93 |
| GLOW | 74.62 | 64.87 | 71.90 | 74.36 |
| | | | | |
| VAE | 191.98 | 155.13 | 154.71 | 156.47 |
| $\beta$-VAE | 112.42 | 93.90 | 92.86 | 93.14 |
| IWAE | 183.04 | 146.70 | 146.49 | 149.16 |

---

[1] https://github.com/mseitzer/pytorch-fid

**Comparisons with the two-step approach in [43].** Manifold overfitting in likelihood-based GMs has been recently analyzed in [43], which provides a two-step procedure to mitigate the issue. The first step maps inputs into a low-dimensional space to handle the intrinsic low dimensionality of the data. This step is then followed by likelihood-based density estimation on the resulting lower-dimensional representation. This is achieved by means of a generalized autoencoder, which relies on a certain set of explicit deep GMs, such as VAEs. Here, we compare our proposal with this two-step approach; results are reported in Table 3. To ensure a fair comparison, we use the same network architecture for our VAE and their generalized autoencoder, and we rely on their official implementation [2]. Following [43], we consider a variety of density estimators in the low-dimensional space such as NFs, Autoregressive Models (ARMs) [77], Adversarial Varitional Bayes (AVB) [46] and Energy-Based Models (EBMs) [16]. We observe that Gaussian mollification is better or comparable with these variants. In addition, our method is extremely simple and readily applicable to any likelihood-based GMs without any extra auxilary models or the need to modify training procedures.

**Table 3:** Comparisons of FID scores on CIFAR10 between mollitication and two-step methods.

| VANILLA VAE | MOLLIFICATION | | TWO-STEP | | | |
|---|---|---|---|---|---|---|
| | VAE+GAUSSIAN | VAE + BLURRING | VAE+NF | VAE+EBM | VAE+AVB | VAE+ARM |
| 191.98 | 155.13 | 175.40 | 208.80 | 166.20 | 153.72 | 203.32 |

## 4.3 Density Estimation on UCI Data Sets

We further evaluate our proposed method in the context of density estimation tasks using UCI data sets [17], following [68]. We consider three different types of normalizing flows: masked autorgressive flows (MAF) [53], REAL-NVP [14] and GLOW [34]. To ensure a fair comparison, we apply the same experimental configurations for both the vanilla and the proposed method, including random seeds, network architectures, optimizer, and the total number of iterations. Table 1 shows the average log-likelihood (the higher the better) on the test data. Error bars correspond to the standard deviation computed over 4 runs. As it can be seen, our proposed Gaussian mollification approach consistently and significantly outperforms vanilla training across all models and all datasets.

**Table 4:** The average test log-likelihood (*higher is better*) on the UCI data sets. Error bars correspond to the standard deviation over 4 runs.

| DATASET | MAF [53] | | REAL-NVP [14] | | GLOW [34] | |
|---|---|---|---|---|---|---|
| | VANILLA | MOLLIFICATION | VANILLA | MOLLIFICATION | VANILLA | MOLLIFICATION |
| RED-WINE | $-16.32 \pm 1.88$ | $-11.51 \pm 0.44$ | $-27.83 \pm 2.56$ | $-12.51 \pm 0.40$ | $-18.21 \pm 1.14$ | $-12.37 \pm 0.33$ |
| WHITE-WINE | $-14.87 \pm 0.24$ | $-11.96 \pm 0.17$ | $-18.34 \pm 2.77$ | $-12.30 \pm 0.16$ | $-15.24 \pm 0.69$ | $-12.44 \pm 0.36$ |
| PARKINSONS | $-8.27 \pm 0.24$ | $-6.17 \pm 0.17$ | $-14.21 \pm 0.97$ | $-7.74 \pm 0.27$ | $-8.29 \pm 1.18$ | $-6.90 \pm 0.24$ |
| MINIBOONE | $-13.03 \pm 0.04$ | $-11.65 \pm 0.09$ | $-20.01 \pm 0.22$ | $-13.96 \pm 0.12$ | $-14.48 \pm 0.10$ | $-13.88 \pm 0.08$ |

**Limitations.** One limitation is that this work focuses exclusively on likelihood-based GMs. On image data, the improvements in FID score indicate that the performance boost is generally substantial, but still far from being comparable with state-of-the-art DMs. While this may give an impression of a low impact, we believe that this work is important in pointing to one of the successful aspects characterizing DMs and show how this can be easily integrated in the optimization of likelihood-based GMs. A second limitation is that, in line with the literature on GMs for image data, where models are extremely costly to train and evaluate, we did not provide error bars on the results reported in the tables in the experimental section. Having said that, the improvements reported in the experiments have been shown on a variety of models and on two popular image data sets. Furthermore, the results are supported theretically and experimentally by a large literature on continuation methods for optimization.

**Broader impact.** This work provides an efficient way to improve a class of GMs. While we focused mostly on images, the proposed method can be applied to other types of data as shown in the density

---

[2] https://github.com/layer6ai-labs/two_step_zoo

estimation experiments on the UCI datasets. Like other works in this literature, the proposed method can have both positive (e.g., synthesizing new data automatically or anomaly detection) and negative (e.g., deep fakes) impacts on society depending on the application.

## 5   Related work

Our work is positioned within the context of improving GMs through the introduction of noise to the data. One popular approach is the use of denoising autoencoders [81], which are trained to reconstruct clean data from noisy samples. Building upon this, [2] proposed a framework for modeling a Markov chain whose stationary distribution approximates the data distribution. In addition, [80] showed a connection between denoising autoencoders and score matching, which is an objective closely related to recent diffusion models [67, 25]. More recently, [45] introduced a two-step approach to improve autoregressive generative models, where a smoothed version of the data is first modeled by adding a fixed level of noise, and then the original data distribution is recovered through an autoregressive denoising model. In a similar vein, [44] recently attempted to use Tweedie's formula [62] as a denosing step, but surprisingly found that it does not improve the performance of NFs and VAEs. Our work is distinct from these approaches in that Gaussian mollification guides the estimated distribution towards the true data distribution in a progressive manner by means of annealing instead of fixing a noise level. Moreover, our approach does not require any explicit denoising step, and it can be applied off-the-shelf to the optimization of any likelihood-based GMs without any modifications.

## 6   Conclusion

Inspired by the enormous success of score-based Diffusion Models (DMs), in this work we hypothesized that data mollification is partially responsible for their impressive performance in generative modeling tasks. In order to test this hypothesis, we introduced data mollification within the optimization of likelihood-based Generative Models (GMs), focusing in particular on Normalizing Flows (NFs) and Variational Autoencoders (VAEs). Data mollification is extremely easy to implement and it has nice theoretical properties due to its connection with continuation methods in the optimization literature, which are well-known techniques to improve optimization. We applied this idea to challenging generative modeling tasks involving imaging data and relatively large-scale architectures as a means to demonstrate systematic gains in performance in various conditions and input dimensions. We measured performance in quality of generated images through the popular FID score.

While we are far from closing the gap with DMs in achieving competing FID score, we are confident that this work will serve as the basis for future works on performance improvements in state-of-the-art models mixing DMs and likelihood-based GMs, and in alternative forms of mollification to improve optimization of state-of-the-art GMs. For example, it would be interesting to study how to apply data mollification to improve the training of GANs; preliminary investigations show that the strategy proposed here does not offer significant performance improvements, and we believe this is due to the fact that data mollification does not help in smoothing the adversarial objective. Also, while our study shows that the data mollification schedule is not critical, it would be interesting to study whether it is possible to derive optimal mollification schedules, taking inspiration, e.g., from [30]. We believe it would also be interesting to consider mixture of likelihood-based GMs to counter problems due to the union of manifolds hypothesis, whereby the intrinsic dimension changes over the input [3]. Finally, it would be interesting to investigate other data, such as 3D point cloud data [85] and extend this work to deal with other tasks, such as supervised learning.

## Acknowledgments and Disclosure of Funding

MF gratefully acknowledges support from the AXA Research Fund and the Agence Nationale de la Recherche (grant ANR-18-CE46-0002 and ANR-19-P3IA-0002).

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

# A  A Primer on Normalizing Flows and VAEs

Given a dataset $\mathcal{D}$ consisting of $N$ i.i.d samples $\mathcal{D} \triangleq \{\mathbf{x}_i\}_{i=1}^N$ with $\mathbf{x}_i \in \mathbb{R}^D$, we aim at estimating the unknown continuous generating distribution $p_{\text{data}}(\mathbf{x})$. In order to do so, we introduce a model $p_{\boldsymbol{\theta}}(\mathbf{x})$ with parameters $\boldsymbol{\theta}$ and attempt to estimate $\boldsymbol{\theta}$ based on the dataset $\mathcal{D}$. A common approach to estimate $\boldsymbol{\theta}$ is to maximize the likelihood of the data, which is equivalent to minimizing the following objective:

$$\mathcal{L}(\boldsymbol{\theta}) \triangleq -\mathbb{E}_{p_{\text{data}}(\mathbf{x})} \left[\log p_{\boldsymbol{\theta}}(\mathbf{x})\right]. \tag{4}$$

Optimization for this objective can be done through a stochastic gradient descent algorithm using minibatches of samples from $p_{\text{data}}(\mathbf{x})$.

## A.1  Normalizing Flows

In flow-based generative models [52, 39], the generative process is defined as:

$$\mathbf{z} \sim p_{\boldsymbol{\phi}}(\mathbf{z}); \quad \mathbf{x} = \mathbf{f}_{\boldsymbol{\psi}}(\mathbf{z}), \tag{5}$$

where $\mathbf{z} \in \mathbb{R}^D$ is a latent variable, and $p_{\boldsymbol{\phi}}(\mathbf{z})$ is a tractable base distribution with parameters $\boldsymbol{\phi}$, such as an isotropic multivariate Gaussian. The function $\mathbf{f}_{\boldsymbol{\psi}} : \mathbb{R}^D \to \mathbb{R}^D$ is invertible, such that given any input vector $\mathbf{x}$ we have $\mathbf{z} = \mathbf{f}_{\boldsymbol{\psi}}^{-1}(\mathbf{x})$. A Normalizing Flow (NF) [59] defines a sequence of invertible transformations $\mathbf{f} = \mathbf{f}_1 \circ \mathbf{f}_2 \circ \cdots \mathbf{f}_K$, such that the relationship between $\mathbf{x}$ and $\mathbf{z}$ can be written as:

$$\mathbf{x} \xleftrightarrow{\mathbf{f}_1} \mathbf{h}_1 \xleftrightarrow{\mathbf{f}_2} \mathbf{h}_2 \cdots \xleftrightarrow{\mathbf{f}_K} \mathbf{z}, \tag{6}$$

where $\mathbf{h}_k = \mathbf{f}_k^{-1}(\mathbf{h}_{k-1}; \boldsymbol{\psi}_k)$ and $\boldsymbol{\psi}_k$ are the parameters of the transformation $\mathbf{f}_k$. For the sake of simplicity, we define $\mathbf{h}_0 \triangleq \mathbf{x}$ and $\mathbf{h}_K \triangleq \mathbf{z}$. The likelihood of the model given a datapoint can be computed analytically using the change of variables as follows:

$$\log p_{\boldsymbol{\theta}}(\mathbf{x}) = \log p_{\boldsymbol{\phi}}(\mathbf{z}) + \log |\det(\partial \mathbf{z}/\partial \mathbf{x})| \tag{7}$$

$$= \log p_{\boldsymbol{\phi}}(\mathbf{z}) + \sum_{k=1}^K \log |\det(\partial \mathbf{h}_k/\partial \mathbf{h}_{k-1})|, \tag{8}$$

where $\log |\det(\partial \mathbf{h}_k/\partial \mathbf{h}_{k-1})|$ is the logarithm of absolute value of the determinant of the Jacobian matrix $\partial \mathbf{h}_k/\partial \mathbf{h}_{k-1}$. This term accounts for the change of measure when going from $\mathbf{h}_{k-1}$ to $\mathbf{h}_k$ using the transformation $\mathbf{f}_k$. The resulting NF model is then characterized by the set of parameters $\boldsymbol{\theta} = \{\boldsymbol{\phi}\} \cup \{\boldsymbol{\psi}_k\}_{k=1}^K$, which can be estimated using the maximum likelihood estimation (MLE) objective Eq. 4.

Though NFs allow for exact likelihood computation, they require $\mathbf{f}_k$ to be invertible and to have a tractable inverse and Jacobian determinant. This restricts the flexibility to certain transformations that can be used within NFs [see e.g., 52, 39, and references therein], such as affine coupling [13, 14], invertible convolution [34], spline [18, 15], or inverse autoregressive transformations [35].

## A.2  Variational Autoencoders

Variational Autoencoders (VAEs) [36, 59] introduce a low-dimensional latent variable $\mathbf{z} \in \mathbb{R}^P$, with $P \ll D$, to the generative process as follows:

$$\mathbf{z} \sim p(\mathbf{z}); \quad \mathbf{x} \sim p_{\boldsymbol{\theta}}(\mathbf{x} \,|\, \mathbf{z}). \tag{9}$$

Here, $p(\mathbf{z})$ is a tractable prior distribution over the latent variables $\mathbf{z}$, and $p_{\boldsymbol{\theta}}(\mathbf{x} \,|\, \mathbf{z})$, which is also known as a decoder, is usually implemented by a flexible neural network parameterized by $\boldsymbol{\theta}$. Different from NFs, VAEs employ a stochastic transformation $p_{\boldsymbol{\theta}}(\mathbf{x} \,|\, \mathbf{z})$ to map $\mathbf{z}$ to $\mathbf{x}$. Indeed, NFs can be viewed as VAEs where the decoder and encoder are modelled by Dirac deltas $p_{\boldsymbol{\theta}}(\mathbf{x} \,|\, \mathbf{z}) = \delta\big(\mathbf{f}_{\boldsymbol{\theta}}(\mathbf{x})\big)$ and $q_{\boldsymbol{\phi}}(\mathbf{z} \,|\, \mathbf{x}) = \delta\big(\mathbf{f}_{\boldsymbol{\theta}}^{-1}(\mathbf{x})\big)$ respectively, using a restricted family of transformations $\mathbf{f}_{\boldsymbol{\theta}}$.

The marginal likelihood of VAEs is intractable and given by:

$$p_{\boldsymbol{\theta}}(\mathbf{x}) = \int p_{\boldsymbol{\theta}}(\mathbf{x} \,|\, \mathbf{z}) p(\mathbf{z}) d\mathbf{z}. \tag{10}$$

A variational lower bound on ther marginal likelihood can be obtained by introducing a variational distribution $q_{\boldsymbol{\phi}}(\mathbf{z} \mid \mathbf{x})$, with parameters $\boldsymbol{\phi}$, which acts as an approximation to the the unknown posterior $p(\mathbf{z} \mid \mathbf{x})$:

$$\log p_{\boldsymbol{\theta}}(\mathbf{x}) \geq \underbrace{\mathbb{E}_{q_{\boldsymbol{\phi}}(\mathbf{z} \mid \mathbf{x})}[\log p_{\boldsymbol{\theta}}(\mathbf{x} \mid \mathbf{z})] - \mathrm{KL}\left[q_{\boldsymbol{\phi}}(\mathbf{z} \mid \mathbf{x}) \parallel p(\mathbf{z})\right]}_{\mathcal{L}_{\mathrm{ELBO}}(\boldsymbol{\theta}, \boldsymbol{\phi})}, \qquad (11)$$

where, $\mathcal{L}_{\mathrm{ELBO}}(\boldsymbol{\theta}, \boldsymbol{\phi})$ is known as the ELBO, and the expectation can be approximated by using Monte Carlo samples from $q_{\boldsymbol{\phi}}(\mathbf{z} \mid \mathbf{x})$. This objective can be optimized with stochastic optimization w.r.t. parameters $\boldsymbol{\theta}$ and $\boldsymbol{\phi}$ in place of Eq. 4.

To tighten the gap between the ELBO and the true marginal likelihood, VAEs can be employed with an expressive form of the approximate posterior $q_{\boldsymbol{\phi}}(\mathbf{z} \mid \mathbf{x})$ such as importance weighted sampling [4] or normalizing flows [35, 79]. In addition, to avoid the over regularization offect induced by the prior $p(\mathbf{z})$, one can utilize a flexible prior such as multi-modal distributions [12, 74], hierarchical forms [66, 38], or simply reweighing the KL divergence term in the ELBO [24].

## B  Details on Blurring Mollification

Recently, [61, 27, 11] have proposed appproaches to destroy information of images using blurring operations for diffusion-type generative models. Their approach involves stochastically reversing the heat equation, which is a PDE that can be used to erase fine-scale information when applied locally to the 2D plane of an image. In particular, the Laplace PDE for heat diffusions is as follows:

$$\frac{\partial}{\partial t}\tilde{\mathbf{x}}(i, j, t) = \Delta\tilde{\mathbf{x}}(i, j, t), \qquad (12)$$

where we consider the initial state of the system to be $\mathbf{x}$, the true image data. This PDE can be effectively solved by employing a diagonal matrix within the frequency domain of the cosine transform, provided that the signal is discretized onto a grid. The solution to this equation at time $t$ can be effectively computed by:

$$\tilde{\mathbf{x}}_t = \mathbf{A}_t\mathbf{x} = \mathbf{V}\mathbf{D}_t\mathbf{V}^{\top}\mathbf{x}, \qquad (13)$$

Here, $\mathbf{V}^{\top}$ and $\mathbf{V}$ denote the discrete cosine transformation (DCT) and inverse DCT, respectively; the diagonal matrix $\mathbf{D}_t$ is the exponent of a weighting matrix for frequencies $\boldsymbol{\Lambda}$ so that $\mathbf{D}_t = \exp(\boldsymbol{\Lambda} t)$. We refer the reader to Appendix A of [61] for the specific definition of $\boldsymbol{\Lambda}$. We can evaluate Eq. 13 in the Fourier domain, which is fast to compute, as the DCT and inverse DCT require $\mathcal{O}(N \log N)$ operations. The equivalent form of Eq. 13 in the Fourier domain is as follows:

$$\tilde{\mathbf{u}}_t = \exp(\boldsymbol{\Lambda} t)\mathbf{u}, \qquad (14)$$

where $\mathbf{u} = \mathbf{V}^{\top}\mathbf{x} = \mathrm{DCT}(\mathbf{x})$. As $\boldsymbol{\Lambda}$ is a diagonal matrix, the above Fourier-space model is fast to evaluate. A Python implementation of this blurring mollification is presented in Algorithm 2.

We follow [61] to set the schedule for the blurring mollification. In particular, we use a logarithmic spacing for the time steps $t_k$, where $t_0 = \sigma_{B,\max}^2/2$ and $t_T = \sigma_{B,\min}^2/2 = 0.5^2/2$, corresponding to sub-pixel-size blurring. Here, $\sigma_{B,\max}^2$ is the effective lengthscale-scale of blurring at the beginning of the mollification process. Following [61], we set this to half the width of the image.

---

**Algorithm 2:** Python code for blurring mollification

```
1  import numpy as np
2  from scipy.fftpack import dct, idct
3
4  def blurring_mollify(x, t):
5      # Assuming the image u is an (KxK) numpy array
6      K = x.shape[-1]
7      freqs = np.pi*np.linspace(0,K-1,K)/K
8      frequencies_squared = freqs[:,None]**2 + freqs[None,:]**2
9      x_proj = dct(u, axis=0, norm='ortho')
10     x_proj = dct(x_proj, axis=1, norm='ortho')
11     x_proj = np.exp(-frequencies_squared * t) * x_proj
12     x_mollified = idct(x_proj, axis=0, norm='ortho')
13     x_mollified = idct(x_mollified, axis=1, norm='ortho')
14     return x_mollified
```

---

# C   Implementation of Noise Schedules

Algorithm 3 shows the Python code for the noise schedules used in this work. For the sigmoid schedule, following [7], we set the default values of `start` and `end` to 0 and 3, respectively.

---

**Algorithm 3:** Python code for noise schedules

---

```python
import numpy as np

def sigmoid(x):
    # Sigmoid function.
    return 1 / (1 + np.exp(-x))

def sigmoid_schedule(t, T, tau=0.7, start=0, end=3, clip_min=1e-9):
    # A scheduling function based on sigmoid function with a temperature tau.
    v_start = sigmoid(start / tau)
    v_end = sigmoid(end / tau)
    return (v_end - sigmoid((t/T * (end - start) + start) / tau)) / (v_end - v_start)

def linear_schedule(t, T):
    # A scheduling function based on linear function.
    return 1 - t/T

def cosine_schedule(t, T, ns=0.0002, ds=0.00025):
    # A scheduling function based on cosine function.
    return np.cos(((t/T + ns) / (1 + ds)) * np.pi / 2)**2
```

---

# D   Experimental Details

## D.1   Data sets

**Synthetic data sets.**

- *Mixture of Gaussians*: We consider a mixture of two Gaussians with means $\mu_k = (2\sin(\pi k), 2\cos(\pi k))$ and covariance matrices $\mathbf{\Sigma}_k = \sigma^2 \mathbf{I}$, where $\sigma = \frac{2}{3}\sin(\pi/2)$. We generate 10K samples for training and 10K samples for testing from this distribution.

- *Von Mises distribution:* We use a von Mises distribution with parameters $\kappa = 1$, and then transform to Cartesian coordinates to obtain a distribution on the unit circle in $\mathbb{R}^2$. We generate 10K training samples and 10K testing from this distribution.

**Image data sets.** We consider two image data sets including CIFAR10 [40] and CELEBA [41]. These data sets are publicly available and widely used in the literature of generative models. We use the official train/val/test splits for both data sets. The resolution of CIFAR10 is $3 \times 32 \times 32$. For CELEBA, we pre-process images by first taking a $148 \times 148$ center crop and then resizing to the $3 \times 64 \times 64$ resolution.

**UCI data sets.** We consider four data sets in the UCI repository [17]: RED-WINE, WHITE-WINE, PARKINSONS, and MINIBOONE. 10% of the data is set aside as a test set, and an additional 10% of the remaining data is used for validation. To standardize the features, we subtract the sample mean from each data point and divide by the sample standard deviation.

## D.2   Software and Computational Resources

We use NVIDIA P100 and A100 GPUs for the experiments, with 16GB and 80GB of memory respectively. All models are trained on a single GPU except for the experiments with NVAE model [78], where we employ two A100 GPUs. We use PyTorch [54] for the implementation of the models and the experiments. Our experiments with VAEs and NFs are relied on the `pythae` [6] and `normflows` [71] libraries, respectively.

### D.3 Training Details

#### D.3.1 Toy examples.

In the experiments on synthetic data sets, we use a REAL-NVP flow [14] with 5 affine coupling layers consisting of 2 hidden layers of 64 units each. We train the model for 20000 itereations using an Adam optimizer [33] with a learning rate of $5 \cdot 10^{-4}$ and a mini-batch size of 256.

#### D.3.2 Imaging experiments.

**REAL-NVP.** We use the multi-scale architecture with deep convolutional residual networks in the coupling layers as described in [14]. For the CIFAR10 data set, we use 4 residual blocks with 32 hidden feature maps for the first coupling layers with checkerboard masking. For the CELEBA data set, 2 resdiual blocks are employed. We use an Adam optimizer [33] with a learning rate of $10^{-3}$ and a mini-batch size of 64. We train the model for 100 and 80 epochs on the CIFAR10 and CELEBA data sets, respectively. For the mollification training, we perturb the data for 50 and 40 epochs for CIFAR10 and CELEBA, respectively.

**GLOW.** We use a multi-scale architecture as described in [34]. The architecture has a depth level of $K = 20$, and a number of levels $L = 3$. We use the AdaMax [33] optimizer with a learning rate of $3 \cdot 10^{-4}$ and a mini-batch size of 64. We train the model for 80 and 40 epochs on the CIFAR10 and CELEBA data sets, respectively. For the mollification training, we perturb the data for 50 and 20 epochs for CIFAR10 and CELEBA, respectively.

**Table 5:** Neural network architectures used for VAEs in our experiments. Here, $\text{CONV}_{(n,s,p)}$ and $\text{CONVT}_{(n,s,p)}$ respectively denotes convolutional layer and transposed convolutional layers with $n$ filters, a stride of $s$ and a padding of $p$, whereas $\text{FC}_n$ represents a fully-connected layer with $n$ units, and BN denotes a batch-normalization layer.

| | CIFAR10 | CELEBA |
|---|---|---|
| ENCODER: | $\mathbf{x} \in \mathbb{R}^{3 \times 32 \times 32}$ | $\mathbf{x} \in \mathbb{R}^{3 \times 64 \times 64}$ |
| | $\to \text{CONV}_{(128,4,2)} \to \text{BN} \to \text{RELU}$ | $\to \text{CONV}_{(128,4,2)} \to \text{BN} \to \text{RELU}$ |
| | $\to \text{CONV}_{(256,4,2)} \to \text{BN} \to \text{RELU}$ | $\to \text{CONV}_{(256,4,2)} \to \text{BN} \to \text{RELU}$ |
| | $\to \text{CONV}_{(512,4,2)} \to \text{BN} \to \text{RELU}$ | $\to \text{CONV}_{(512,4,2)} \to \text{BN} \to \text{RELU}$ |
| | $\to \text{CONV}_{(1024,4,2)} \to \text{BN} \to \text{RELU}$ | $\to \text{CONV}_{(1024,4,2)} \to \text{BN} \to \text{RELU}$ |
| | $\to \text{FLATTEN} \to \text{FC}_{256 \times 2}$ | $\to \text{FLATTEN} \to \text{FC}_{256 \times 2}$ |
| DECODER: | $\mathbf{z} \in \mathbb{R}^{256} \to \text{FC}_{8 \times 8 \times 1024}$ | $\mathbf{z} \in \mathbb{R}^{256} \to \text{FC}_{8 \times 8 \times 1024}$ |
| | $\to \text{CONVT}_{(512,4,2)} \to \text{BN} \to \text{RELU}$ | $\to \text{CONVT}_{(512,5,2)} \to \text{BN} \to \text{RELU}$ |
| | $\to \text{CONVT}_{(256,4,2)} \to \text{BN} \to \text{RELU}$ | $\to \text{CONVT}_{(256,5,2)} \to \text{BN} \to \text{RELU}$ |
| | $\to \text{CONVT}_{(3,4,1)}$ | $\to \text{CONVT}_{(128,5,2)} \to \text{BN} \to \text{RELU}$ |
| | | $\to \text{CONVT}_{(3,4,1)}$ |

**VAEs.** We use convolutional networks for both the encoder and decoder of VAEs [36, 60]. Table 5 shows the details of the network architectures. We use an Adam optimizer [33] with a learning rate of $3 \cdot 10^{-4}$ and a mini-batch size of 128. We train the model for 200 and 100 epochs on the CIFAR10 and CELEBA data sets, respectively. For the mollification training, we perturb the data for 100 and 50 epochs for CIFAR10 and CELEBA, respectively. The addtional details of the variants of VAEs are as follows:

- VAE-IAF [35]: We use a 3-layer MADE [20] with 128 hidden units and RELU activation for each layer and stack 2 blocks of Masked Autoregressive Flow to create the flow for approximating the posterior.

- $\beta$-VAE [24]: We use a coefficient of $\beta = 0.1$ for the Kullback-Leibler divergence (KL) term in the ELBO objective.

- IWAE [4]: We use a number of importance samples of $K = 5$.

- HVAE [5]: We set the number of leapfrog steps to used in the integrator to 1. The leapfrog step size is adaptive with an initial value of $0.001$

**NVAE.** We use the default network architecture as described in [78]. We train the model on the CIFAR10 for 300 epochs with an AdaMax optimizer [33] with a learning rate of $10^{-3}$ and a mini-batch size of 200. For the mollification training, we perturb the data for first 150 epochs.

## D.4 UCI experiments

For MAF models [53], we employ 5 autoregressive layers, each composed of a feedforward neural network utilizing masked weight matrices [20]. This neural networks consists of 512 hidden units and employ the tanh activation function. For REAL-NVP [14] and GLOW [34] models, we implement 5 coupling layers, each comprising two feedforward neural networks with 512 hidden units to handle the scaling and shift functions. The first neural network utilizes the tanh activation function, while the latter employs a rectified linear activation function. We introduce batch normalization [29] after each coupling layer in REAL-NVP and after each autoregressive layer in MAF. All models are trained with the Adam optimizer [33] for 150 epochs with a learning rate of $10^{-4}$ and a mini-batch size of 100. For mollification training, we employ Gaussian mollification during the entire training process.

## D.5 Evaluation Metrics

**Maximum Mean Discrepancy.** The MMD between two distributions $p_{\text{data}}$ and $p_{\boldsymbol{\theta}}$ is defined as follows [21]:

$$\text{MMD}(p_{\text{data}}, p_{\boldsymbol{\theta}}) = \sup_{\|h\|_{\mathcal{H}} \leq 1} \Big[ \mathbb{E}_{p_{\text{data}}}[h] - \mathbb{E}_{p_{\boldsymbol{\theta}}}[h] \Big], \tag{15}$$

where $\mathcal{H}$ denotes a reproducing kernel Hilbert space (RKHS) induced by a characteristic kernel $K$. The MMD has the closed form:

$$\text{MMD}^2(p_{\text{data}}, p_{\boldsymbol{\theta}}) = \mathbb{E}_{\mathbf{x}, \mathbf{x}' \sim p_{\text{data}}}[K(\mathbf{x}, \mathbf{x}')] + \mathbb{E}_{\mathbf{x}, \mathbf{x}' \sim p_{\boldsymbol{\theta}}}[K(\mathbf{x}, \mathbf{x}')] - 2\mathbb{E}_{\mathbf{x} \sim p_{\text{data}}, \mathbf{x}' \sim p_{\boldsymbol{\theta}}}[K(\mathbf{x}, \mathbf{x}')], \tag{16}$$

which can be estimated by using samples from $p_{\text{data}}$ and $p_{\boldsymbol{\theta}}$. In our experiments, we use 10K test samples from the true data distribution $p_{\text{data}}$ and 10K samples from the model distribution $p_{\boldsymbol{\theta}}$ for estimating the MMD score.

We employ the popular RBF kernel for the MMD, which is defined as follows:

$$K(\mathbf{x}, \mathbf{x}') = \sigma^2 \exp\Big( -\frac{\|\mathbf{x} - \mathbf{x}'\|^2}{2l} \Big), \tag{17}$$

with a lengthscale $l = 1$ and variance $\sigma^2 = 10^{-4}$.

**FID score.** To assess the quality of the generated images, we employed the widely used Fréchet Inception Distance [23]. The FID measures the Fréchet distance between two multivariate Gaussian distributions, one representing the generated samples and the other representing the real data samples. By comparing their distribution statistics, we can assess the similarity between the generated and real data distributions. The FID score is defined as follows:

$$\text{FID} = \|\mu_{\text{real}} - \mu_{\text{gen}}\|^2 + \text{Tr}(\Sigma_{\text{real}} + \Sigma_{\text{gen}} - 2\sqrt{\Sigma_{\text{real}}\Sigma_{\text{gen}}}). \tag{18}$$

The distribution statistics are obtained from the 2048-dimensional activations of the pool3 layer of an Inception-v3 network. We use the `pytorch-fid` [3] library for calculating the FID score in our experiments.

# E  Addtional Results

Table 6 and Table 7 illustrate uncurated samples from the trained models. Fig. 9 and Fig. 10 show the progression of FID scores during training on the CIFAR10 and CELEBA datasets, respectively.

---

[3] https://github.com/mseitzer/pytorch-fid

**Table 6:** Uncurated samples from the models trained on the CIFAR10 dataset.

| | VANILLA | GAUSS. MOLLIFICATION | BLUR. MOLLIFICATION |
|---|---|---|---|
| REAL-NVP | | | |
| GLOW | | | |
| VAE | | | |
| VAE-IAF | | | |
| IWAE | | | |
| $\beta$-VAE | | | |
| HVAE | | | |

**Table 7:** Uncurated samples from the models trained on the CELEBA dataset.

|  | VANILLA | GAUSS. MOLLIFICATION | BLUR. MOLLIFICATION |
|---|---|---|---|
| REAL-NVP | | | |
| GLOW | | | |
| VAE | | | |
| VAE-IAF | | | |
| IWAE | | | |
| $\beta$-VAE | | | |
| HVAE | | | |

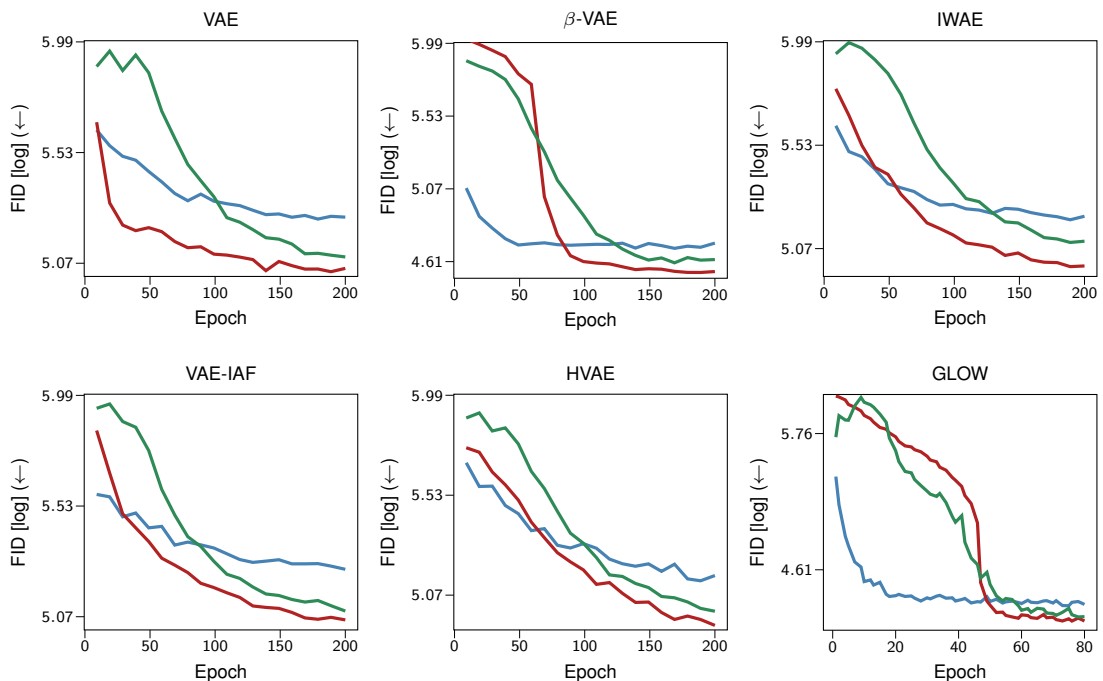

**Figure 9:** The progression of FID scores during training on the CIFAR10 dataset.

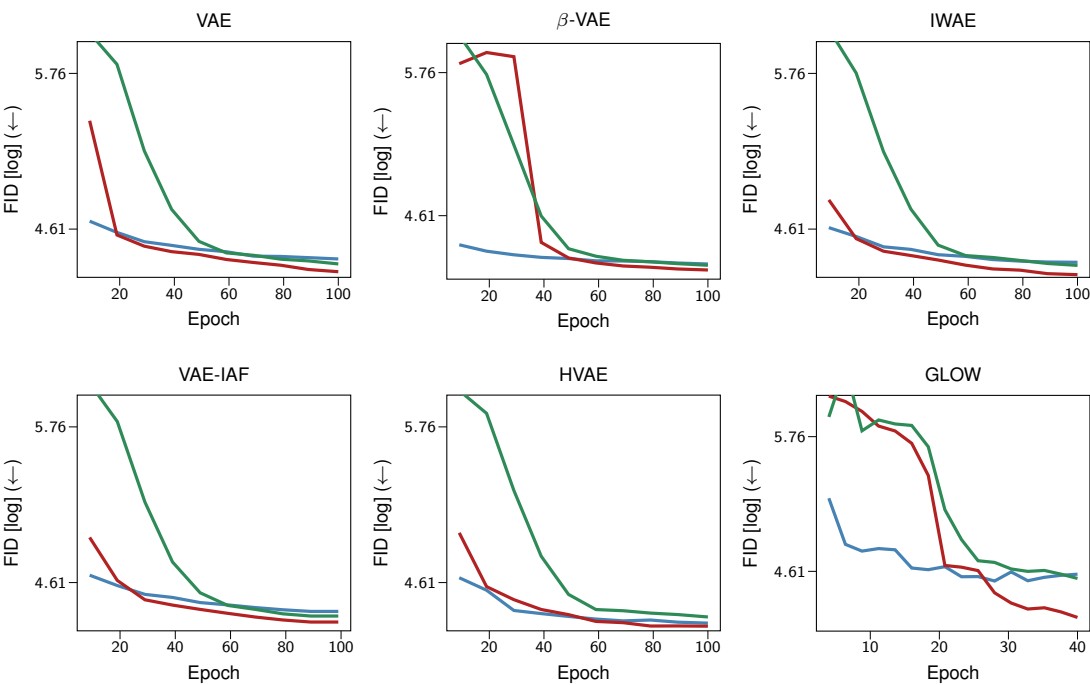

**Figure 10:** The progression of FID scores during training on the CELEBA dataset.

