# OpenReview forum: "One-Line-of-Code Data Mollification Improves Optimization of Likelihood-based Generative Models"
_NeurIPS.cc/2023/Conference — NeurIPS 2023 poster_

### Official Review · Reviewer_sZKb · 2023-07-05

**Soundness:** 3 good
**Presentation:** 3 good
**Contribution:** 2 fair
**Rating:** 6
**Confidence:** 4

**Summary:**

This paper proposes the use of data mollification to improve sample quality from likelihood-based generative models. The authors observe that likelihood-based models are worse-performing than score-based models due to manifold overfitting (i.e., learning the manifold but not the distribution on it), and poor density estimation in low-density regions (demonstrated with toy examples). Data mollification is proposed as a solution to these issues and demonstrated visually with toy mathematical constructions. On CIFAR10 and CelebA (64x64), the methodology is shown to improve FID scores and there is a visible qualitative improvement in the model samples. Different choices of mollification processes and annealing schedules are explored and compared.

**Strengths:**

- Main results show significant improvements in FID score on multiple image datasets by introducing a technique that entails no additional computational cost. These hold across different image datasets. Samples are not cherrypicked.
- Results are positive on two families of generative models, normalizing flows and variational autoencoders.
- The considered hypotheses suggesting why likelihood-based models suffer from sample quality are backed with theoretical work and experiments with toy examples, where visualization of these phenomena can be confirmed.
- Various hyperparameters (e.g., the choices of mollification process and schedule) are compared, and the main results are ablated against the same model without the proposed methodology. This gives confidence in their methodology truly demonstrating improvement on the datasets and models considered.
- The paper is very well written, has clear and easy-to-follow language, and the literature review is more than satisfactory. There are a few issues on presentation (see questions section in my review).

**Weaknesses:**

- My biggest concern is that the image datasets considered might be too simple to conclude that these techniques will materialize into improvement for larger-scale datasets and models. One potential way to do this is to study, e.g., ImageNet 64x64; usually, positive results there are much more likely to hold in larger scales due to the large scale and difficulty of ImageNet itself. Conversely, results in datasets with very narrow distributions (low entropy) like CIFAR10 and CelebA might not transfer to very diverse, more challenging datasets in practice. Studying the effect of the methodology on more challenging datasets will make the results much more valuable and significant.

- The methodology is not very substantial due to its extreme simplicity, nor is mollification new to the literature as acknowledged by the authors themselves. Arguably, however, applying mollification to the data in a schedule for generative models is not something done in practice for generative models, and contrary to score-based models, it does not require learning all noise levels, so it could be seen as a clever and creative way of applying similar ideas from the literature. Simplicity can be positive, and perception of novelty can be extremely subjective, so I am not too concerned about this. Still, due to how short the contribution is, exploring more models would been very welcome since the paper is generally about likelihood-based generative models.

- Minor: while there is enough visual and empirical evidence, there is little discussion for *why* data mollification alleviates the problems of manifold overfitting and accurate density estimation in low-density regions. Since there is a lot of discussion and even theorems from prior work in the body of the paper, a theoretical lens, or at least hypotheses or discussion around this question feel lacking and would be very welcome.

**Questions:**

- The single most important detail: can the authors confirm that the *targets* of the loss are mollified? I am assuming that this is indeed the case as they mention that mollification == smoothening the loss landscape, but this is not explicitly stated in the main body of the paper, and a reader might interpet the methodology as a certain kind of data augmentation where only the input is degraded via noise/blurring/etc in a schedule. I strongly suggest reflecting this in Algorithm 1.
- Are models trained with the vanilla approach trained for the same amount of steps as those with data mollification? It might be possible that the models have different convergence speeds because the data distribution changes. It would be nice to show that the strongest baselines are considered in terms of training both approaches with the best possible number of steps.
- Can we include samples for at least one image dataset in the body of the paper? All metrics are flawed, and visualizing the samples themselves is key to observing what the better numbers actually translate to (also, FID scores are not comparable between datasets, so it is hard to judge what the quantiative improvements entail without looking at samples).
- Nit: can we drop the use of "ablation" for *comparisons* of different schedules, and refer to the vanilla models as an ablation study? Ablation == removing a component.

**Limitations:**

As mentioned in the weaknesses section, the biggest limitation is that the datasets considered might be too narrow, and there is uncertainty on the methodology being helpful for larger / higher entropy / more diverse datasets . Utilizing larger-scale data essential for larger scale models to attain good results, and it has been the case in many prior work that, e.g., results in CIFAR10 can differ vastly to more realistic image datasets. In my opinion, addressing this successfully should flip this paper to acceptance.

Another limitation is that, as the authors note, score-based models (which are also likelihood-based generative models-- they can be interpreted as hierarchical VAEs with a fixed approximate posterior) are state-of-the-art image generators but are not considered in experiments. For smaller-scale tasks (e.g. CIFAR10) or the toy examples, this could have been possible without requiring significant computational resources. It is possible that naive gaussian mollification of the data might interfere with a score model's ability to learn the noise level adequately, so there is some uncertainty that it will simply work as intended on this model family which is dominating the field. Moreover, score-based models have an extremely similar coarse-to-fine inductive bias: it is possible that the same benefit provided by data mollification is already provided (albeit at sampling time) and, consequently, the methodology might not improve score-based models at all. Including these results could increase the value of this work significantly if there are positive results, and make (1) the impact of the work more explicit and (2) the exploration thorough and up to date with the field, even if results there are not good. Flows and VAEs are both very relevant to the field and there is a demonstrated contribution in these; more than one model family, and the existing findings are still potentially valuable to the community.

As noted by the authors, experiments also are limited to mathematical constructions and to image datasets. Focusing a paper on the image modality is reasonable, so I am not concerned about this. Addressing the other main limitations in my opinion will increase the value of the contributions sufficiently for this paper to merit acceptance.

---

> ### Author Rebuttal · Authors · 2023-08-09
>
> **Q15. My biggest concern is that the image datasets considered might be too simple ... One potential way to do this is to study, e.g., ImageNet 64x64...**
>
> Thank you for your comment. Unfortunately, prior to submission, we faced limitations in compute capacity, preventing us from exploring such a dataset that demands significant training efforts. In addition, due to the restricted time frame of the rebuttal phase, we are unable to conduct experiments on the ImageNet 64x64 dataset. However, we respectfully disagree that the CIFAR10 and CelebA datasets are too simple. These datasets are widely recognized as standard benchmarks for evaluating generative models. It is worth noting that recent works addressing manifold-overfitting challenges in generative models have also utilized these datasets. For instance, Loaiza-Ganem et al. (2022) employed much simpler datasets such as MNIST, FMNIST, SVHN. Additionally, Meng et al. (2021) considered the same datasets but focused solely on a single autoregressive model, PixelCNN++. In contrast, our approach involved rigorous testing across a diverse set of VAEs and normalizing flows, showcasing consistent and significant improvements in all experiments. Therefore, we believe that our empirical evidence is strong enough to demonstrate the effectiveness of our proposed data mollification method.
>
> Ref:
>
> -  Loaiza-Ganem et al. Diagnosing and Fixing Manifold Overfitting in Deep Generative Models. TMLR 2022.
> - Meng et al. Improved Autoregressive Modeling with Distribution Smoothing. ICLR 2021 Oral.
>
>
> **Q16. Minor: while there is enough visual and empirical evidence, there is little discussion for why data mollification alleviates the problems of manifold overfitting and accurate density estimation in low-density regions...**
>
> Thank you for your comment. Our mollification approach is theoretically motivated by Theorem 1. Indeed, we also further discuss this theorem from line 166 to line 185.
>
> **Q17. Can the authors confirm that the targets of the loss are mollified? I am assuming that this is indeed the case as they mention that mollification == smoothening the loss landscape, but this is not explicitly stated in the main body of the paper ... I strongly suggest reflecting this in Algorithm 1.**
>
> Thanks for your comment. We confirm that the targets of the loss are mollified. In fact, in the Introduction (line 66) we clarified that *“Strictly speaking, data mollification in score-based DMs and likelihood-based GMs are slightly different. In the latter, the amount of noise injected in the data is continuously annealed throughout training. At the beginning, the equivalent loss landscape seen by the optimizer is much smoother, due to the heavy perturbation of the data, and a continuous reduction of the noise level allows optimization to be gracefully guided until the point where the level of noise is zero”*. We appreciate the reviewer's suggestion and we will further elaborate on this when discussing Algorithm 1 in the camera-ready.
>
> **Q18. Are models trained with the vanilla approach trained for the same amount of steps as those with data mollification? It might be possible that the models have different convergence speeds because the data distribution changes...**
>
> Thanks for your comment. Indeed, we explicitly stated in line 259 that *“Nevertheless, to ensure a  fair comparison, we adopt identical settings for the vanilla and the proposed approaches, including random seed, optimizer, and the total number of iterations.”* Furthermore, we also report the progression of FID scores during training in Figure 2 of the Appendix.
>
> **Q19. Can we include samples for at least one image dataset in the body of the paper? All metrics are flawed ... so it is hard to judge what the quantiative improvements entail without looking at samples).**
>
> Thanks for your comment. Although the metrics we used may be flawed, they are widely accepted as standard ways to assess performance of generative models in the literature. However, we agree with your point and will move some qualitative samples from the Appendix to the main paper of the camera ready.
>
> **Q20. Nit: can we drop the use of "ablation" for comparisons of different schedules, and refer to the vanilla models as an ablation study?...**
>
> Thanks for your comment. We will drop the use of “ablation” for comparisons of different schedules in the camera ready.
>
> **Q21. Another limitation is that, as the authors note, score-based models ... are state-of-the-art image generators but are not considered in experiments ... Moreover, score-based models have an extremely similar coarse-to-fine inductive bias: it is possible that the same benefit provided by data mollification is already provided (albeit at sampling time) and, consequently, the methodology might not improve score-based models ...**
>
> Thanks for your very interesting comment. As the Reviewer mentioned, we believe that our data mollification might not improve score-based models as these models are already equipped with a data mollification mechanism. Our primary objective is not to improve score-based diffusion models. Instead, as stated in line 47, our paper aims to address the question: *Can we leverage the strengths of score-based diffusion models to enhance likelihood-based generative models without incurring the high cost of sample generation?*. Furthermore, our work proves beneficial in untangling the diverse factors contributing to the exceptional performance of score-based diffusion models. Specifically, we hypothesize that data mollification serves as a distinctive element that sets these models apart from other likelihood-based models, such as VAEs and normalizing flows.

---

> > ### Comment · Reviewer_sZKb · 2023-08-13
> > **I have read the rebuttal**
> >
> > Thanks for the rebuttal. I will increase my score to 6. After reading it, I agree with the authors that the approach is theoretically motivated by the mathematical presentation, and that even though score-based models are not within the scope of the paper, there is value in both (A) the improved results for other model families and (B) in understanding the success of score-based models further by studying other models in isolation. Note that, jut like how I asked for clarification of the targets being mollified in my review, this was a point of confusion for other reviewers as well. It would be helpful to add a bold header "Differences between data mollification and score-based models" or something of the sort where it is discussed in the paper to make this as explicit as possible.
> >
> > I still believe that this work would be significantly stronger with more empirical results (which could be either more difficult image datasets, likelihood-based models from prior work that have performance such as VDVAE (Child, 2020),  or some other kind of modality). However, as I mentioned in my review, I already believe that the amount of experiments is acceptable. Hence my final score of 6 but not higher.

---

> > > ### Author Response · Authors · 2023-08-16
> > > **Response to Reviewer sZKb**
> > >
> > > We thank the reviewer for the reply to our rebuttal and the positive feedback.
> > >
> > > > It would be helpful to add a bold header "Differences between data mollification and score-based models" or something of the sort where it is discussed in the paper to make this as explicit as possible.
> > >
> > > Thank you for your comment. We will further highlight the differences between our proposed data mollification and score-based models in the camera-ready
> > >
> > > > I still believe that this work would be significantly stronger with more empirical results (which could be either more difficult image datasets … or **some other kind of modality**). However, as I mentioned in my review, I already believe that the amount of experiments is acceptable
> > >
> > > Thank you for your positive comment. We have managed to conduct a new set of experiments with different types of data. In particular, inspired by the work of Song and Ermon (2020), we consider four datasets in the UCI machine learning repository for the task of density estimation. We test our mollification procedure against vanilla training using three different types of normalizing flows including masked autoregressive flow (MAF), realNVP, and Glow.  We adopt identical settings for the vanilla and our method, including random seeds, network architectures, optimizer, and the total number of iterations.
> > >
> > > The table below shows the average log-likelihood (*the higher the better*) on the test data. Error bars correspond to the standard deviation computed over 4 runs. As it can be seen, our data mollification approach consistently and significantly outperforms vanilla training across all models and all datasets. This new empirical evidence further confirms the effectiveness of our method.
> > >
> > > |            | MAF - Vanilla |   MAF - Mollif.   | RealNVP - Vanilla | RealNVP - Mollif. | Glow - Vanilla |   Glow - Mollif.  |
> > > |------------|:-------------:|:-----------------:|:-----------------:|:-----------------:|:--------------:|:-----------------:|
> > > | RedWine    | -16.32 ± 1.88 | **-11.51 ± 0.44** |   -27.83 ± 2.56   | **-12.51 ± 0.40** |  -18.21 ± 1.14 | **-12.37 ± 0.33** |
> > > | WhiteWine  | -14.87 ± 0.24 | **-11.96 ± 0.17** |   -18.34 ± 2.77   | **-12.30 ± 0.16** |  -15.24 ± 0.69 | **-12.44 ± 0.36** |
> > > | Parkinsons |  -8.27 ± 0.24 |  **-6.17 ± 0.17** |   -14.21 ± 0.97   |  **-7.74 ± 0.27** |  -8.29 ± 1.18  |  **-6.90 ± 0.24** |
> > > | Miniboone  |  -13.03 ± .04 | **-11.65 ± 0.09** |   -20.01 ± 0.22   | **-13.96 ± 0.12** |  -14.48 ± 0.10 | **-13.88 ± 0.08** |
> > >
> > > We thank the Reviewer for this comment, which ultimately strengthens even more the message of our paper. We will include these new results in the camera-ready. We would be happy to engage in  any follow-up discussion or address any additional comments.
> > >
> > > Ref:
> > > - Song and Ermon. Bridging the Gap Between f-GANs and Wasserstein GANs. ICML 2020.

---

### Official Review · Reviewer_qdL5 · 2023-07-05

**Soundness:** 2 fair
**Presentation:** 2 fair
**Contribution:** 2 fair
**Rating:** 4
**Confidence:** 4

**Summary:**

In this paper, the authors address the limitation of likelihood-based Generative Models (GMs) in achieving high-quality samples compared to state-of-the-art score-based Diffusion Models (DMs). They propose a novel approach to enhance the generation quality of GMs by incorporating data mollification, a technique used in score-based DMs for accurate density estimation in low-density regions and to mitigate manifold overfitting.

The authors connect data mollification with Gaussian homotopy, a well-known optimization technique, by adding Gaussian noise to the data. This simple addition can be implemented with just one line of code in the optimization loop of likelihood-based GMs. Through experiments on popular image datasets using various likelihood-based GMs, such as variational autoencoders and normalizing flows, the authors demonstrate improvements in generation quality, as measured by the FID score.

Notably, this approach provides a boost in generation quality without incurring additional computational overheads. This work could potentially be a step towards bridging the gap between likelihood-based GMs and score-based DMs, offering new possibilities for generating high-quality samples in various domains.

**Strengths:**

The approach presented is both straightforward and seamlessly integrable into the training process of various generative models. The authors provide lucid instructions and accompanying pseudo code for its implementation. Notably, the results displayed in table 1 exhibit a remarkable enhancement compared to the baseline performance.

**Weaknesses:**

The approach described in the paper lacks explicit discussion regarding its relationship to noise augmentation. However, towards the end of section 3, it appears that the authors have chosen a temperature factor of 0.7, which leads to a substantial mollification towards the end of the training process. Additionally, their results are comparable to what is typically observed with conventional data augmentation techniques.

Given that the evaluation of the presented method in this paper is entirely based on empirical analysis, it is crucial for the authors to empirically demonstrate that their approach surpasses alternative methods. Replacing the sigmoid noise schedule with a uniform/constant function would be the equivalent to simple data augmentation with Gaussian noise. Since data augmentation using Gaussian noise is a common practice in the training of many generative models, it becomes essential to provide empirical evidence showcasing that the proposed noise schedule yields results that go beyond what can be achieved with a uniform noise schedule. The authors should provide results from experiments with different amounts of noise and show that their noise schedule outperforms all of these.

The authors present little theoretical evidence for the choice of noise function. They justify their choice of the sigmoid function by referencing Jabri et al., who demonstrated improved stability with the sigmoid function compared to the cosine function through empirical results. However, the empirical findings displayed in table 2 contradict this justification, indicating that a cosine noise schedule performs better than the sigmoid noise schedule in all cases except GLOW.

**Questions:**

1. What is the relationship between the described approach and noise augmentation?
2. How does applying different amounts of gaussian noise continuously (uniform noise schedule) to the data compare to the sigmoid function?
3. How does the presented method outperform alternative approaches that address manifold overfitting, excluding data augmentation?
4. Why was the sigmoid function selected over the cosine function given the results shown in table 2?
It is not clear if the procedure mentioned on line 258 is applied to all training or only certain models or dataset. What is the role of this on the presented FID numbers in table 1?

**Limitations:**

1. Lack of explicit discussion on the relationship to noise augmentation: The paper does not provide a thorough examination or explanation of how the approach relates to noise augmentation. While a temperature factor of 0.7 is mentioned, there is no detailed discussion on the significance of this factor or its impact.
2. Exclucive reliance on empirical analysis: The evaluation of the presented method relies solely on empirical analysis. To establish the superiority of their approach, the authors should conduct comparative experiments with alternative (much simpler) methods and demonstrate that their approach consistently outperforms them. Specifically, they should compare their noise schedule against different amounts of constant noise and show its superiority over a set of variations.
3. Limited theoretical justification for the choice of noise function: The paper lacks substantial theoretical evidence supporting the selection of the sigmoid noise function. Although the authors refer to Jabri et al.'s empirical results to justify their choice, the findings presented in table 2 contradict this justification by showing that a cosine noise schedule performs better in all cases except GLOW.

---

> ### Author Rebuttal · Authors · 2023-08-09
>
> **Q11. The approach described in the paper lacks explicit discussion regarding its relationship to noise augmentation. However, towards the end of section 3, it appears that the authors have chosen a temperature factor of 0.7, which leads to a substantial mollification towards the end of the training process.**
>
> We believe that there is a fundamental misunderstanding here. We explicitly explained in line 195 that our approach involves creating a sequence of progressively less smoothed versions of the original data, rather than applying substantial mollification towards the end of the training process. The visualization in Figure 3 of the main paper provides a clear illustration of the sigmoid schedule with different temperatures. As depicted, the noise variance $\sigma^2_t$ gradually decreases from 1 to 0 as the iteration $t$ increases, across all values of the temperature hyper-parameter.
>
> **Q12. Given that the evaluation of the presented method in this paper is entirely based on empirical analysis, it is crucial for the authors to empirically demonstrate that their approach surpasses alternative methods. Replacing the sigmoid noise schedule with a uniform/constant function would be the equivalent to simple data augmentation with Gaussian noise... it becomes essential to provide empirical evidence showcasing that the proposed noise schedule yields results that go beyond what can be achieved with a uniform noise schedule. The authors should provide results from experiments with different amounts of noise and show that their noise schedule outperforms all of these ...**
>
> Thanks for your comment. Indeed, we already discussed carefully the approach of adding noise into the data for generative models in the related work section, from line 317: *“Our work is positioned within the context of improving GMs through the introduction of noise to the data…”*. Recent works [34,36,35] showed that naively adding a fixed amount of noise to the data can degrade the quality of generated samples as the model is completely trained on the modified data, not with the true data distribution. Our work is distinct from these approaches in that Gaussian mollification guides the estimated distribution towards the true data distribution in a progressive manner by means of annealing instead of fixing a noise level. This is inspired by the reverse process of diffusion models. As the Reviewer suggested, we have performed a new set of experiments with the suggested data augmentation using Gaussian noise (uniform noise schedule). We follow the suggestion in [36] to set the noise variance $\sigma^2=0.3$ and also consider a smaller noise, i.e. $\sigma^2=0.1$. The results of the synthetic mixture of Gaussians and von Mise distribution and on CIFAR-10 are attached in the general response. We can see that the uniform noise schedule consistently performs worse than our proposed approach of annealing noise using the sigmoid function.
>
> **Q13. The authors present little theoretical evidence for the choice of noise function. They justify their choice of the sigmoid function by referencing Jabri et al., who demonstrated improved stability with the sigmoid function compared to the cosine function through empirical results. However, the empirical findings displayed in table 2 contradict this justification, indicating that a cosine noise schedule performs better than the sigmoid noise schedule in all cases except GLOW ... Why was the sigmoid function selected over the cosine function  ... It is not clear if the procedure mentioned on line 258 is applied to all training or only certain models or dataset. What is the role of this on the presented FID numbers in table 1?.... While a temperature factor of 0.7 is mentioned, there is no detailed discussion on the significance of this factor or its impact.**
>
> Thanks for your comment. In response to Q5 of Reviewer yxxe's, we thoroughly discussed both the theoretical motivation and empirical evidence guiding our choice of noise schedule.
>
> - As presented in Table 2, the sigmoid function demonstrates comparable performance to the cosine function in most cases, with only marginal differences favoring the cosine function. However, notably, the sigmoid function outperforms the cosine function significantly for the Glow model. Based on these observations, we have decided to adopt the sigmoid function as the default noise schedule for our experiments.
>
> - The temperature of the sigmoid function is a hyper-parameter, and ideally should be chosen for different datasets [6].  This hyper-parameter governs the rate of data mollification, controlling how rapidly the noise variance diminishes. After extensive experimentation, we found that a temperature factor of 0.7 consistently yields favorable outcomes across all experiments, making it a sensible default value.
>
> - We confirm that the procedure mentioned in line 258 is consistently applied to all experiments with image data. We also further elaborate on this in our response to Reviewer yxxe's question (Q6).
>
> **Q14. How does the presented method outperform alternative approaches that address manifold overfitting, excluding data augmentation? … Exclucive reliance on empirical analysis: The evaluation of the presented method relies solely on empirical analysis. To establish the superiority of their approach, the authors should conduct comparative experiments with alternative (much simpler) methods and demonstrate that their approach consistently outperforms them.**
>
> Thanks for your comment. We believe that the reviewer potentially missed our experiments to compare our approach with a two-stage procedure proposed by [34] in mitigating manifold overfitting. We found that our proposed Gaussian mollification is better or comparable with this approach.  In addition, our method is extremely simple and readily applicable to any likelihood-based generative models without any extra auxiliary models or the need to heavily modify training procedures.

---

> > ### Comment · Reviewer_qdL5 · 2023-08-14
> >
> > Thank you for the rebuttal.
> >
> > Q11. Let me provide further clarification: The selected temperature factor will result in significant mollification beyond t > 0.5. Even though the mollification at the final t=T might be zero, its implications on the optimized model remain uncertain. This is due to the substantial mollification occurring over a significant portion of the training process and shortly before the training concludes.
> >
> > Q12. Thanks you for running the additional tests. Interestingly, for all cases except Glow **there was a significant improvement with a uniform noise schedule**. Why is this?
> > Note: **The authors have tuned the temperature factor, so for this to be comparable, the sigma in the uniform noise schedule should be tuned aswell.**
> >
> > I have raised my score to 4.

---

> > > ### Author Response · Authors · 2023-08-16
> > > **Response to Reviewer qdL5 (1)**
> > >
> > > We thank the reviewer for the reply to our rebuttal.
> > >
> > > > Q11. Let me provide further clarification: The selected temperature factor will result in significant mollification beyond t > 0.5. Even though the mollification at the final t=T might be zero, its implications on the optimized model remain uncertain. This is due to the substantial mollification occurring over a significant portion of the training process and shortly before the training concludes.
> > >
> > > Thank you for providing additional clarification. As addressed in our response to *Q5 of Reviewer yxxe*, the selected temperature results in a noise schedule that decreases steeply during the initial stages of training. This aligns with recent results for diffusion models (Chen, 2023; Hoogeboom et al., 2023). This is reasonable for our method as we expect more iterations for lower noise to capture finer data distribution details, and fewer for higher noise with predominantly Gaussian mollified data for easier learning.
> > >
> > > Moreover, as we discussed in response to *Q6 of Reviewer yxxe*,  for complicated datasets like high-dimensional images, we apply data mollification during the first half of the optimization phase, and we continue optimizing the model using the original data in the second half. We hypothesize that, in the first phase, our data-mollification procedure gradually steers the model optimization towards global optima regions. Consequently, we observed that for the image experiments, conducting further training of the models using the true distribution resulted in enhanced convergence to global optima, leading to improvement in image quality. Notice that this strategy is consistently applied to all experiments with image data.
> > >
> > > Ref:
> > > - Chen. On the Importance of Noise Scheduling for Diffusion Models. Arxiv 2023.
> > > - Hoogeboom et al. simple diffusion: End-to-end Diffusion for High Resolution Images. ICML 2023.
> > >
> > > [Please see the next box due to character limits]

---

> > > ### Author Response · Authors · 2023-08-16
> > > **Response to Reviewer qdL5 (2)**
> > >
> > > > Q12. Thanks you for running the additional tests. Interestingly, for all cases except Glow there was a significant improvement with a uniform noise schedule. Why is this? Note: The authors have tuned the temperature factor, so for this to be comparable, the sigma in the uniform noise schedule should be tuned aswell.
> > >
> > > Thanks for your further comment. First of all, we would like to recall the motivation of our proposed method that aims to address two challenges: (1) density estimation in low-density regions and (2) manifold overfitting. By adding Gaussian noise to the data, we effectively increase the intrinsic dimension of the high-dimensional data. Thus, this mechanism helps counteract manifold overfitting. We discussed this in the main paper and in response to *Q10 of Reviewer yxxe*. This is arguably why data augmentation with Gaussian noise (using a uniform noise schedule) could potentially enhance the performance of likelihood generative models.
> > >
> > > However, as we demonstrated in experiments with the synthetic Gaussian-mixture dataset in the paper and in the rebuttal, adding a small amount of noise can reduce manifold overfitting but cannot help the model effectively estimate the density in low-density regions. To address this, a Gaussian noise with a large variance spanning the entire input space is required. However, given the substantial modification introduced to the original data distribution, such an approach can significantly compromise the quality of generated samples. This was demonstrated by Loaiza-Ganem et al. (2021) and Meng et al. (2022).
> > >
> > > In this work, instead of employing a fixed amount of noise, we progressively introduce multiple levels of noise during the training, guided by Theorem 1. Initially, our mollification process considers a simplified, coarse-grained version of the target density, spanning the full input space. Subsequently, the method gradually reduces the level of noise, allowing for progressive refinement of the estimated versions of the target density. This procedure shows remarkable results in our experiments.
> > >
> > > Nevertheless, as suggested by the Reviewer, we have performed a new set of experiments to do a grid search of the noise variance for the uniform schedule. The FID results (*the lower the better*) on CIFAR10 dataset are shown in the below table. As can be seen, by tuning a suitable noise variance for the uniform schedule, the result of Glow is improved. The results also indicate that the noise variance for the uniform schedule is highly model-dependent. Meanwhile, our proposed data mollification technique, employing the sigmoid schedule, still consistently outperforms the alternatives.
> > >
> > > Furthermore, it is worth mentioning that our proposed method is quite robust with the choice of noise schedule, as *we used a sigmoid schedule with a temperature of 0.7 across all experiments, showcasing remarkable and consistent improvements*. This characteristic further shows a distinctive advantage of our method, bypassing the necessity of fine-tuning the noise variance for the uniform schedule.
> > >
> > > |   *FID score*          | Vanilla | Sigmoid (Ours) | Uniform ($\sigma^2 = 0.3$) | Uniform ($\sigma^2 = 0.1$) | Uniform ($\sigma^2 = 0.03$) | Uniform ($\sigma^2 = 0.01$) | Uniform ($\sigma^2 = 0.003$) | Uniform ($\sigma^2 = 0.001$) |
> > > |-------------|:-------:|:-------:|:--------------------------:|:--------------------------:|:---------------------------:|:---------------------------:|:----------------------------:|:----------------------------:|
> > > | RealNVP     |  191.98 |  121.75 |           192.45           |           131.47           |            153.82           |            128.17           |            135.81            |            131.04            |
> > > | Glow        |  74.62  |  64.87  |           305.71           |           204.29           |            207.67           |            248.77           |             138.7            |            167.40            |
> > > | VAE         |  191.98 |  155.13 |           160.12           |           161.93           |            161.79           |            162.93           |            167.75            |            166.54            |
> > > | $\beta$-VAE |  112.42 |  93.90  |           133.04           |            96.01           |            100.40           |           95.2390           |             96.67            |             99.54            |
> > >
> > > We would like to thank you again for your review and further feedback. We would be happy to do any follow-up discussion or address any additional comments.
> > >
> > > Ref:
> > > - Loaiza-Ganem et al. Diagnosing and Fixing Manifold Overfitting in Deep Generative Models. TMLR 2022.
> > > - Meng et al. Improved Autoregressive Modeling with Distribution Smoothing. ICLR 2021 Oral.

---

### Official Review · Reviewer_yxxe · 2023-07-07

**Soundness:** 3 good
**Presentation:** 4 excellent
**Contribution:** 2 fair
**Rating:** 6
**Confidence:** 4

**Summary:**

The paper proposes an optimization method that improves the training of likelihood-based generative models. The central idea is that in the early phases of the optimization, the learner tries to model a smoothed version of the likelihood function. As the optimization proceeds, the smoothening decreases (and the difficulty of the optimization problem increases) with the end goal of learning the likelihood of the true distribution. The authors show that this simple idea (inspired by the diffusion models literature) gives consistent improvements in the performance of likelihood-based generative models.

**Strengths:**

The authors borrow an idea from the diffusion literature (the mollification of the data distribution at many levels) and they apply it in an interesting way to likelihood-based models such as VAEs and Normalizing Flows. Interestingly, the modeling target changes during the optimization (starting from easy distributions and slowly going to harder ones). Instead, diffusion models try to model all the distributions simultaneously. As far as I know, this method has not been proposed in previous works and it seems to be effective in improving likelihood-based models.

The paper is clearly written and the experimental methodology seems valid.

The authors showcase the effectiveness of their method starting from informative toy examples and moving to real datasets (CIFAR-10 and CelebA). The boost in performance is consistent across the experiments.


**Weaknesses:**

Even though the method increases the effectiveness of likelihood-based models, the performance is still very weak. The authors acknowledge this limitation. This is not necessarily an important weakness since future work might further improve the performance of likelihood-based models to the point they become competitive with diffusion models. However, I do not see how this particular optimization method could be improved to achieve these results.

I do not see why the noise schedule has to be similar to the noise schedule diffusion models use. In the design process of a diffusion model, we typically select the corruption scheduling such that the difficulty of transitioning from one corruption level to the next one during sampling is roughly the same across all corruption levels. Why does this intuition transfer here on how we select the scheduling?

I am also puzzled why there is no catastrophic forgetting happening. The authors mention that for the last 50% of the optimization, they stop using their method of mollification and they train for the true distribution.








**Questions:**

I would love to hear what the authors think about directions for future work to improve the method and further enhance the performance of likelihood-based models.

Related question, do the authors see any way to trade sampling speed for performance using some modification of their method?

For the scheduling, would it be possible to move to the next level once the loss flattens? This seems to me a reasonable thing to do in order to move from easier optimization problems to hard ones, using the solution of the previous problem as a warm start. I have seen this trick in the Intermediate Layer Optimization work in which the goal is again to move gradually from easier optimization problems to harder ones.

I think I missed the point about blurring in the paper. Intuitively, it seems to me that both blurring and the addition of noise have the same goal of making the distribution "easier". Diffusion models are based on the idea of sampling from hard distributions starting with a warm start from an easier distribution. Is it possible that some other scheduling of the blurring kernel would lead to better results? How about a combination of blurring and noise as in Soft Diffusion?


**Limitations:**

In my opinion, the main limitations of the paper are: i) the poor understanding of why catastrophic forgetting is not happening and how to select the scheduling of the corruption, ii) the absence of a way to trade sampling speed with quality. For ii) the FID numbers right now are very weak. This is not necessarily a limitation specific only to this work, but a limitation of one-step sampling models in general.

---

> ### Author Rebuttal · Authors · 2023-08-09
>
> **Q4. ... I do not see how this particular optimization method could be improved to achieve these results.**
>
> We agree with Reviewer that our method does not necessarily help likelihood-based generative models **(GMs)** reach the performance level of diffusion models **(DMs)**, as this heavily relies on the nature of these models. However, our results clearly demonstrates that our method consistently enhances the performance of likelihood-based GMs. The key advantage lies in the simplicity and ease of applicability of our approach, which can be readily integrated into any likelihood-based GMs without incurring additional costs or necessitating model modifications. As a result, our work holds valuable practical implications for practitioners and may open up new avenues for exploring and enhancing likelihood-based GMs (see Q7).
>
> **Q5. I do not see why the noise schedule has to be similar to the noise schedule diffusion models use...**
>
> Our noise schedule choice is guided by Theorem 1, advocating diminishing Gaussian noise. Any monotonically decreasing function from 1 to 0 is a vaid noise schedule. This property is similar to noise schedules used for DMs. However, we agree that the noise schedule for our method is not necessarily the same as that for DMs. Notice that the choice of noise schedule for DMs is also mainly heuristic and empirical-based [39,20,6].  We drew inspiration from noise schedules in the diffusion literature and carefully evaluated them. The sigmoid schedule showed remarkable results across all our experiments. This aligns with recent results for DMs showing that the noise distribution skewed towards noisier levels is more helpful [6,20]. This is reasonable for our method as we expect more iterations for lower noise to capture finer data distribution details, fewer for higher noise with predominantly Gaussian mollified data for easier learning.
>
> **Q6. ... why there is no catastrophic forgetting happening...**
>
> We hypothesize that our data-mollification procedure gradually steers the model optimization towards global optima regions, as illustrated in Figure 6. Consequently, we observed that for the image experiments, conducting further training of the models using the true distribution resulted in enhanced convergence to global optima, leading to improvement in image quality. To establish a relevant connection, we associate this procedure with Gaussian homotopy, a widely recognized technique for non-convex optimization. For an example and further insights, you can refer to [16], particularly Figure 3 in that reference, which provides a helpful illustration.
>
> **Q7. ... what the authors think about directions for future work ...**
>
> In Conclusion, we outlined several directions for future work aimed at enhancing our method and improving the performance of likelihood-based models. Specifically, in line 349, we mentioned that *“...derive optimal mollification schedules ... consider mixtures of likelihood-based GMs to counter problems due to the union of manifolds hypothesis...”*. Additionally, we believe it would be worthwhile to consider an extension of our method that accommodates multiple noise levels simultaneously, similar to the concept seen in DMs. Recent work on Gaussian homotopy for optimization (Lin et al., 2023) could serve as a source of inspiration for exploring this direction and potentially yielding even better results.
>
> Ref: Lin et al. Continuation Path Learning for Homotopy Optimization. ICML 2023 Oral.
>
> **Q8. ... Do the authors see any way to trade sampling speed for performance using some modification of their method?...**
>
> We believe that there is a possible misunderstanding. Indeed, our method does not require modifying the nature of likelihood-based GMs, but only requires a small modification in the training loop as shown in Algorithm 1. Throughout our experiments, we have focused on NFs and VAEs, which are one-step sampling GMs. As we mention in the response to Q7, an interesting direction is to extend our method to consider multiple levels of noise taking inspiration from a recent work on continual path learning for homotopy optimization (Lin et al., 2023).  By adopting this approach, we can efficiently trade training time to achieve a more optimal solution.
>
> **Q9. Would it be possible to move to the next level once the loss flattens?...**
>
> We agree with Reviewer that designing an adaptive noise schedule using training information might be efficient. While we have not yet explored this approach because our proposed method operates under stochastic optimization, we acknowledge its potential and find it promising for enhancing our method's effectiveness further. We will include this direction in the Conclusion.
>
> **Q10. I think I missed the point about blurring...**
>
> We explored blurring as an alternative method to perturb image data, starting from an easier distribution. However, our experiments showed that Gaussian mollification outperforms blurring and, in some cases, blurring even harms the generative performance when compared to vanilla training (Table 1). One of the main motivations behind our proposed Gaussian mollification is to address manifold overfitting. By adding Gaussian noise to the data, we effectively increase the intrinsic dimension of the high-dimensional data [34,36]. Thus, this mechanism helps prevent manifold overfitting. Meanwhile, the blurring process averages out the images in the dataset, resulting in a contraction of the data into an even lower-dimensional subspace [48]. For example, at the first iteration, the effective dimensionality is reduced to one. This contraction could potentially magnify manifold overfitting, explaining why Gaussian mollification performs better than blurring. These findings are consistent with results observed in DMs [48]. Given these insights, we believe that a combination of blurring and noise, as seen in Soft Diffusion, could lead to improved results compared to solely relying on blurring.

---

> > ### Comment · Reviewer_yxxe · 2023-08-12
> > **Response to Rebuttal**
> >
> > Thank you for your rebuttal. I will increase my score to 6.

---

### Official Review · Reviewer_S7wY · 2023-07-27

**Soundness:** 3 good
**Presentation:** 3 good
**Contribution:** 1 poor
**Rating:** 3
**Confidence:** 4

**Summary:**

The paper proposes a idea to borrow one of the strengths of score-based DMs, which is the ability to perform accurate density estimation in low-density regions and to address manifold overfitting by means of data mollification. We connect data mollification through the addition of Gaussian noise to Gaussian homotopy, which is a well-known technique to improve optimization. Data mollification can be implemented by adding one line of code in the optimization loop, and the authors demonstrate that this provides a boost in generation quality of likelihood-based GMs, without computational overheads.

**Strengths:**

The paper presents a fairly easy to use data mollification techniques to help vae and flow model to generate better samples, especially for low data density region.

**Weaknesses:**

1.The first concern of this paper is its novelty and it is very critical for conference like Neurips. It basically states the same thing as in "Generative modeling by estimating gradients of the data distribution", a NeurIPS 2019 oral paper. Due to the identical proposition and analysis of generative model, the novel element in this paper is very limited. The difference is instead of inference the score, the paper just let the generative networks to train under the mollified data.

2. The description of the model is vague, except for algorithm1, there is no clear presentation of how to use the model. Although in line 235, it says "This process uses the solution from one level of mollification as a means to guiding optimization for the next.", do the authors suggest, in a generative task (lets say image generation) a sample is generated from white noise by following the gradient of each gaussian level? The connection between langevine dynamics, SDE, diffusion models are very strong to this kind of sampling.


**Questions:**

1. What's the difference in terms of motivation between the famous "Generative modeling by estimating gradients of the data distribution" paper and this one?

2. What differentiates this paper from SDE score-matching and langevine dynamics generative models?

3. How to combine this model with VAE? Any clear description or examples?

**Limitations:**

1. The model as described, seems to make the training process very heavy and slow, similar to diffusion models.

---

> ### Author Rebuttal · Authors · 2023-08-09
>
> **Q1. The first concern of this paper is its novelty and it is very critical for conference like Neurips. It basically states the same thing as in "Generative modeling by estimating gradients of the data distribution", a NeurIPS 2019 oral paper. Due to the identical proposition and analysis of generative model, the novel element in this paper is very limited. The difference is instead of inference the score, the paper just let the generative networks to train under the mollified data … What's the difference in terms of motivation between the famous "Generative modeling by estimating gradients of the data distribution" paper and this one?**
>
> Thanks for your comment. Indeed, our work is motivated by the success of score-based models and inspired by Song et al. 2019.  We emphasize this multiple times in the paper, for example from line 118 to line 128. In this work, we identify that one of the key elements making score-based diffusion models so performant is the data mollification mechanism as mentioned by Song et al. 2019. However, other likelihood-based models are not equipped with this mechanism. The key research question we aim to address in this paper is whether or not the data mollification mechanism can also be helpful for other likelihood-based generative models and how to implement this procedure effectively. Our work is theoretically backed and provides extensive empirical evidence. Other Reviewers acknowledge this novelty. For example, Reviewer sZKb commented that *“applying mollification to the data in a schedule for generative models is not something done in practice for generative models, and contrary to score-based models, it does not require learning all noise levels, so it could be seen as a clever and creative way of applying similar ideas from the literature”*. We respect the position of the Reviewer on the novelty of our work; however, we believe that science progresses through the integration of existing knowledge. There are many great works built up or combined from existing works. We believe that our results will be valuable to the scientific community and practitioners.
>
> **Q2. The description of the model is vague, except for algorithm1, there is no clear presentation of how to use the model. Although in line 235, it says "This process uses the solution from one level of mollification as a means to guiding optimization for the next.", do the authors suggest, in a generative task (lets say image generation) a sample is generated from white noise by following the gradient of each gaussian level? The connection between langevine dynamics, SDE, diffusion models are very strong to this kind of sampling…. What differentiates this paper from SDE score-matching and langevine dynamics generative models?**
>
> Thanks for your comment. We strongly believe that there is a fundamental misunderstanding here. Indeed, in this work, we do not propose any model but a method to mollify data during the training to improve likelihood-based generative models. The key difference between this work and “SDE score-matching and Langevin dynamics generative models” is that we do not consider these models. Instead, we borrow one of the key elements of these models, which is the data mollification mechanism, to apply other likelihood-based models, such as VAEs and Normalizing Flows. Notice that these models are one-step sampling models, and we do not change the nature of these models but only change the training procedure as described clearly from line 195 to line 208, and Algorithm 1. In addition, we provide a theoretical justification and extensive empirical evidence to demonstrate the effectiveness of our proposed method.
>
> **Q3. The model as described, seems to make the training process very heavy and slow, similar to diffusion models…..How to combine this model with VAE? Any clear description or examples?**
>
> Thanks for your comment. We respectfully disagree with this point as our method does not introduce any additional computational overheads to the training pipeline of likelihood-based models. This is one of the main advantages of our method and is acknowledged by other Reviewers. This comment makes us believe that there is a fundamental misunderstanding of our paper by the Reviewer. Algorithm 1 clearly describes how to integrate our mollification procedure into any training loop of likelihood-based models like VAEs and normalizing flows. Basically, we only introduce a simple additional step into the training loop to mollify the data according to a noise schedule. Appendix C clearly provides the Python code for how to implement this noise schedule.

---

> ### Comment · Area_Chair_eA2m · 2023-08-21
>
> The authors have provided a rebuttal which appears to address or contradict a number of your criticisms.  Please be sure to read over these points carefully and provide a response.  Your full participation in the review process is critical for its success.  Thank you!

---

### Author Rebuttal · Authors · 2023-08-09

Dear Reviewers,

We thank the Reviewers for careful reviews of our paper and for insightful comments, which have helped us improve the paper significantly. We are encouraged by the endorsements in the initial reviews that: 1) **the paper is very well written** (Reviewer sZKb and yxxe); 2) **the proposed method is novel, straightforward and clever, and has not been proposed in previous works and it seems to be effective in improving likelihood-based models** (Reviewer sZKb, qdL5 and yxxe); 3) **the experimental methodology is valid** (Reviewer yxxe) and **the results are positive** (Reviewer sZKb, qdL5); 4) **the literature review is more than satisfactory** (Reviewer sZKb).

However, there are potential misunderstandings and concerns from the reviewers, particularly Reviewer S7wY and qdL5. We believe that we have been able to address the Reviewers' comments by clarifying certain sections of the paper and incorporating some additional experiments. Details on these changes can be found in the response to individual comments by the Reviewers. The attached file contains additional results used to address Q12 of Reviewer qdL5.

---

### Decision · Program_Chairs · 2023-09-21

**Decision:**

Accept (poster)

**Comment:**

The reviewers identified some issues and areas of confusion.  During the discussion phase the authors provided clarifications to these issues which led to reviewers increasing their scores.  As a result and after having read the reviews, discussion and paper I recommend acceptance.